# SALS: Sparse Attention in Latent Space for KV cache Compression

**Junlin Mu**[1,2,3],[*] **Hantao Huang**[3],[†] **Jihang Zhang**[3], **Minghui Yu**[3],[†] **Tao Wang**[1,2],[†] **Yidong Li**[1,2],[†]
[1]Key Laboratory of Big Data & Artificial Intelligence in Transportation, Ministry of Education, China
[2]School of Computer Science & Technology, Beijing Jiaotong University, China
[3]ByteDance Seed, China

## Abstract

Large Language Models (LLMs) capable of handling extended contexts are in high demand, yet their inference remains challenging due to substantial Key-Value (KV) cache size and high memory bandwidth requirements. Previous research has demonstrated that KV cache exhibits low-rank characteristics within the hidden dimension, suggesting the potential for effective compression. However, due to the widely adopted Rotary Position Embedding (RoPE) mechanism in modern LLMs, naive low-rank compression suffers severe accuracy degradation or creates a new speed bottleneck, as the low-rank cache must first be reconstructed in order to apply RoPE. In this paper, we introduce two key insights: first, the application of RoPE to the key vectors increases their variance, which in turn results in a higher rank; second, after the key vectors are transformed into the latent space, they largely maintain their representation across most layers. Based on these insights, we propose the Sparse Attention in Latent Space (SALS) framework. SALS projects the KV cache into a compact latent space via low-rank projection, and performs sparse token selection using RoPE-free query–key interactions in this space. By reconstructing only a small subset of important tokens, it avoids the overhead of full KV cache reconstruction. We comprehensively evaluate SALS on various tasks using two large-scale models: LLaMA2-7b-chat and Mistral-7b, and additionally verify its scalability on the RULER-128k benchmark with LLaMA3.1-8B-Instruct. Experimental results demonstrate that SALS achieves SOTA performance by maintaining competitive accuracy. Under different settings, SALS achieves 6.4-fold KV cache compression and 5.7-fold speed-up in the attention operator compared to FlashAttention2 on the 4K sequence. For the end-to-end throughput performance, we achieves 1.4-fold and 4.5-fold improvement compared to GPT-fast on 4k and 32K sequences, respectively. The source code will be publicly available in the future.

## 1 Introduction

The groundbreaking success of Large Language Models (LLMs), such as ChatGPT [15] and Claude [1], is transforming information retrieval in many areas. The exponential growth in LLM service requests is creating an unprecedented demand for inference optimization algorithms, especially for long-context applications. In particular, many research efforts [14, 20, 28, 33] show that the Key-Value cache (KV cache) acts as a major performance bottleneck in LLM serving. As the sequence length increases, the KV cache consumes a large portion of GPU device memory. This

---

[*]Work done at ByteDance
[†]Corresponding authors.twang@bjtu.edu.cn,{huanghantao,yuminghui.exp}@bytedance.com,ydli@bjtu.edu.cn

39th Conference on Neural Information Processing Systems (NeurIPS 2025).

Table 1: KV data movement, memory cost and complexity comparison of quantization, low rank and token sparse method that is with fixed, dynamic and local token selection strategy.

| Name | Methods | KV data movement | Memory size | Computation Complexity | Accuracy |
|---|---|---|---|---|---|
| Palu [4] | Low Rank | Median | Low | High | Low |
| Loki [20] | Dynamic + Low Rank | Low | Median | Median | Median |
| StreamingLLM [33] | Fixed pattern | Low | Median | Low | Low |
| Quest [22] | Dynamic | Low | High | Median | Median |
| DS [29] | Dynamic | Low | Median | Median | High |
| Hshare [25] | Dynamic | Low | Median | Median | High |
| **SALS (Ours)** | Dynamic+Low Rank | Low | Low | Midian | High |

high resource requirement underscores the urgent need to mitigate the KV cache memory overhead and further improve attention efficiency.

To tackle the KV cache overhead challenge, previous works [4, 20] show that the KV cache exhibits low-rank characteristics within the hidden dimension, and thus can be effectively compressed in the latent space. Palu [4] further shows that there exists a clear trade-off between accuracy and reconstruction overhead for multi-head attention. Compressing each KV head separately reduces the overhead but at the cost of losing accuracy. However, compressing all heads together by low-rank projection improves accuracy by preserving the global information but results in significantly increased reconstruction cost. In this work, we show that sparsely selecting a small subset of tokens from the KV cache significantly reduces reconstruction error while preserving model accuracy.

Recently, Rotary Position Embedding (RoPE) [21] has been widely adopted in LLMs; it introduces sinusoidal positional information by multiplying query and key states with rotation matrices. This not only prevents the fusion of low rank weights into the query state, but also requires the reconstruction of key states from the latent space. In this work, we observe that the application of RoPE to the key vectors increases their variance, which in turn results in a higher rank. Therefore, the KV cache must be compressed before applying RoPE, which leads to high reconstruction complexity. We further observe that the key vectors in the latent space largely preserve their representation across most layers. Inspired by Double Sparse [29], we use these compressed key vectors as token selection guidance, so that only the selected tokens need to be reconstructed, which significantly reduces the reconstruction complexity.

In this paper, we propose the Sparse Attention in Latent Space (SALS) framework. SALS projects all attention heads into a shared single-head latent space via low-rank projection, for both pre-RoPE queries and pre-RoPE keys. The approximation attention scores are computed in the latent space and the top-k tokens are selected accordingly. These selected tokens are then reconstructed and applied with RoPE for the final attention computation. By reconstructing only a small subset of all tokens, it avoids the overhead of full KV cache reconstruction.

In summary this paper makes the contributions as follow:

- We observe that applying RoPE to key vectors leads to increased variance, resulting in higher rank and lower compression rates. In , key vectors before RoPE can still maintain the representation for the critical token selection.

- We propose the Sparse Attention in Latent Space (SALS) framework to utilize the low-rank pre-ROPE KV cache compression and further to adopt the KV cache in latent space to select critical tokens. The sparse attention is then proposed with reduced KV data movement and computational complexity.

- We comprehensively evaluate SALS on various tasks using two large-scale models: LLaMA2-7b-chat and Mistral-7b, and additionally verify its scalability on the RULER-128k benchmark with LLaMA3.1-8B-Instruct. Experimental results show that we can achieves much higher accuracy comparing to the low-rank based KV cache compression. Comparing to sparse attention works, under the similar benchmark accuracy, SALS achieves 6.4-fold KV cache compression and 5.7-fold speed-up in attention operator compared to FlashAttention2 on 4K sequence. For the end-to-end throughput performance, SALS achieves 1.4-fold and 4.5-fold improvement compared to GPT-fast on 4k and 32K sequences respectively.

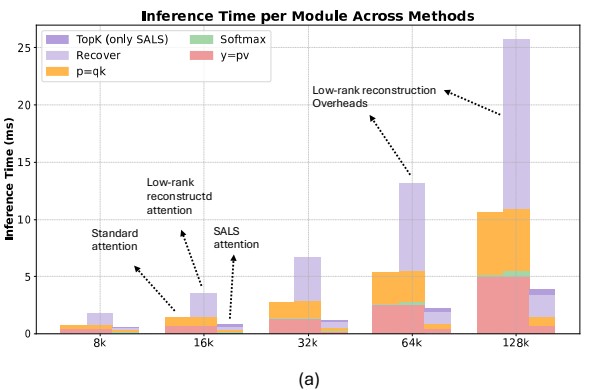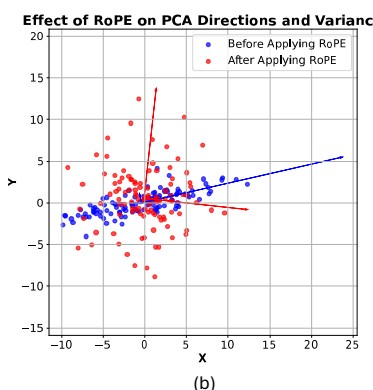

Figure 1: (a) Increasing inference time due to low-rank matrix reconstruction. Low-rank KV cache with overhead leads to longer inference time than the standard attention due to the reconstruction overhead; (b) A simplifed key vector example with changed PCA direction after applying RoPE

## 2 Related work

### 2.1 Attention in Latent Space

The low rank matrix decomposition such as singular value decomposition (SVD) to compress LLMs has been actively studied. Existing works such as ASVD [31], SVD-LLM [24] and CALDERA [18] are proposed to compress LLM weights. However, as the context sequence length and batch size increase, the KV cache size is getting larger than LLM weights, urging the need for KV cache compression. Eigen Attention [19] has been proposed to compress the KV cache after applying RoPE but suffers a relatively large accuracy loss. To improve the accuracy, the most naive way is to compress the pre-RoPE KV cache. However, as Palu [4] points out, this will greatly introduce additional computation for recovering the key vectors. Palu further proposes the grouped-head low-rank decomposition optimization to reduce the computation overhead. We agree with Palu's observation but tackle the computation from the sparse perspective. By only selecting a subset of all the tokens (the critical tokens), the low-rank key vectors reconstruction complexity is greatly reduced. By integrating sparse attention, we observe that the KV cache size, data movement and computation complexity are all reduced as summarized in Table 1.

### 2.2 Sparse Attention

Early efforts to reduce attention complexity in transformers, such as Sparse Transformer [5], Reformer [12], and Longformer [3], primarily focused on training-based approaches to achieve sparse attention. In contrast, recent studies [17, 22, 33] have explored *post-training sparse attention*, leveraging the inherent sparsity of attention scores to identify the most salient tokens without retraining. For instance, StreamingLLM [26] identifies initial and recent tokens as critical, while others [32, 33] accumulate attention scores to locate important tokens in the KV cache. Quest [22] assesses KV cache page importance via min/max metrics, and Double-sparse [29] selects key KV tokens across important channels. However, most of these post-training approaches mainly reduce bandwidth and computation cost, leaving the KV cache uncompressed and thus a potential bottleneck [20, 28].

Natively Sparse Attention (NSA) [30] introduces a dynamic hierarchical sparse strategy combining coarse-grained token compression with fine-grained token selection. Unlike SALS, NSA requires training from scratch to learn a compressed attention module. In contrast, SALS can be calibrated *post-training*, performing top-$k$ token selection based on low-rank latent approximations rather than a trained sparse module. This allows SALS to achieve comparable sparsity behavior without retraining and with precise control over the KV-cache compression process.

Overall, our work addresses the limitation of prior sparse attention methods by storing and operating on a low-rank KV cache, then selectively reconstructing high-dimensional representations for exact sparse attention. This design preserves accuracy while substantially reducing memory and computation overhead.

# 3 Challenges on Latent Space Transformation

In this section, we first review the RoPE mechanism preventing the fusion of query and key states in latent space. We also show that after RoPE, the key vector variance increases as well as the rank. We then analyze the pre-RoPE representational capability of tokens in the latent space, focusing on their ability to preserve key semantic features.

## 3.1 Latent Space Transformation with RoPE

The introduction of the RoPE mechanism complicates the low-rank transformation into the latent space. We begin by considering the case without RoPE. Let $\mathbf{K} \in \mathbb{R}^{s \times d}$ denote the key matrix, where $s$ is the number of tokens and $d$ is the dimensionality of each head. By applying an orthonormal projection matrix $\mathbf{U} \in \mathbb{R}^{d \times r}$, with $r \ll d$, we can transforme origin Key matrix into latent space:

$$\widetilde{\mathbf{K}} = \mathbf{K}\,\mathbf{U} \tag{1}$$

so that each original Key vector $\mathbf{k}_t \in \mathbb{R}^d$ is mapped to latent space $\widetilde{\mathbf{k}}_t \in \mathbb{R}^r$ leading to kv cache compression rate $d/r$. This method works well on the basis of that

$$\mathbf{U}\mathbf{U^T} \approx \mathbf{I}, \;\; \mathbf{Q}\mathbf{K^T} \approx \mathbf{Q}\mathbf{U}\mathbf{U^T}\mathbf{K^T} \tag{2}$$

However, once RoPE is introduced, the relative order between the RoPE rotation matrix $\mathbf{R}$ and the lowrank projector $\mathbf{U}\mathbf{U}^\top$ becomes crucial. In Equation 3, we show both design choices in a single expression: (i) pre-RoPE transformation, where keys are first transformed and then rotated, and (ii) post-RoPE transformation, where keys are rotated first and transformed afterwards. For a query at position $i$ and a key at position $j$, the attention score be approximated in two lowrank forms:

$$\mathbf{Q}_i\,\mathbf{K}_j^\top \;\approx\; \underbrace{\mathbf{Q}\,\mathbf{R}_i\big(\mathbf{R}_j^\top\,\mathbf{U}\mathbf{U}^\top\big)\mathbf{K}^\top}_{\text{pre-RoPE}} \quad \text{or} \quad \underbrace{\mathbf{Q}\,\mathbf{R}_i\big(\mathbf{U}\mathbf{U}^\top\,\mathbf{R}_j^\top\big)\mathbf{K}^\top}_{\text{post-RoPE}}, \tag{3}$$

where $\mathbf{R}_i$ and $\mathbf{R}_j$ are the RoPE rotation matrices for positions $i$ and $j$, respectively.

Ideally, we can adopt the post-RoPE transformation to cache $\widetilde{\mathbf{K}} = \mathbf{K}\mathbf{R}_j\mathbf{U}$ and transform the rotated query into a latent space $\widetilde{\mathbf{Q}} = \mathbf{Q}\mathbf{R}_i\mathbf{U}$ to avoid the reconstruction complexity. However, as we will discuss later, applying RoPE will rotate the key vectors with larger variance, making them difficult to approximate using a low-rank matrix. On the other hand, in the pre-RoPE transformation, caching $\widetilde{\mathbf{K}}$ and reconstructing the full-rank keys as $\widetilde{\mathbf{K}}\mathbf{U}^\top$ will result in substantial overhead.

**Variance amplification for post–RoPE rotation.** Previous works [4, 20] observe that the key cache exhibits low-rank characteristics with a relatively small rank to retain approximately 90% of the energy. As such, we can use a single projection weight $\mathbf{U}$ as shown in Equation 1 to compress the key cache. However, we find that this phenomenon only holds for the key cache prior to applying RoPE. From Figure 1(b), we observe that applying RoPE to a set of key vectors pushes the data points outward and rotates their principal axes. As shown in Figure 1(b), the principal component is rotates away from its original direction and the points become more scattered with two main principal components. This indicates the increased rank due to applying RoPE. Moreover, the principal components rotate based on the token position, suggesting that a single fixed projection matrix may no longer approximate all of these rotated subspaces. Therefore, a pre–RoPE key for latent space transformation is preferable for maintaining accuracy.

**Compute overhead during pre–RoPE reconstruction** From Figure 1(a), we compare full attention with pre–RoPE low–rank compression across sequence lengths ranging from 1 K to 32 K tokens. While low–rank compression can reduce memory size, the time spent reconstructing low-rank keys/values from their rank-$r$ form and applying the RoPE rotation grows linearly with sequence length and soon dominates the total attention runtime at 32 K tokens. This aligns with our analysis of Equation 3, which intruoduces a trade-off of memory savings and computational cost.

## 3.2 Token Representation in Latent Space

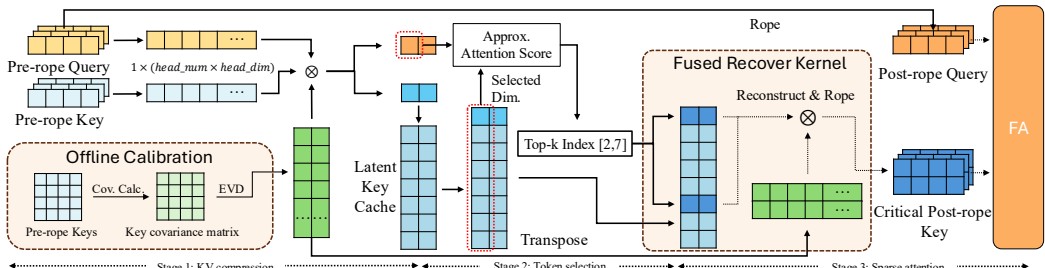

Figure 3: Overall architecture of SALS. Three stages are introduced with stage 1 for multi-head KV Cache compression, stage 2 for token selection in latent space and stage 3 for sparse attention.

To quantify the token representations of the query and key in the latent space, we first compute the latent-space attention vector $\widetilde{\mathbf{p}} = \widetilde{\mathbf{Q}}\widetilde{\mathbf{K}}$, where in the pre-RoPE setting, $\widetilde{\mathbf{Q}} = \mathbf{Q}\mathbf{U}$. Let $\mathcal{C} = \{i_1, i_2, \ldots, i_{N_c}\} \subseteq \{1, \ldots, s\}$, $N_c \ll s$, denote the index set corresponding to the top-$N_c$ entries of $\widetilde{\mathbf{p}}$. The *overlap score (OS)* measures the token representation in the latent space, which is fraction of the full attention mass that these indices capture. $OS = \sum_{i \in \mathcal{C}} p_i / \sum_{i=1}^{s} p_i$, where $p_i$ denotes the $i$-th entry of the full attention distribution $p \in \mathbf{R}^s$.

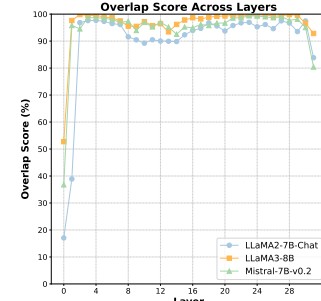

Figure 2: High overalp rate of pre-RoPE across layers

Figure 2 presents the overlap score for LLaMA–7B-chat, LLaMA–8B and Mistral-7B-v0.2. We observe that the average overlap score mostly remains above $90\%$ for layers 2-29 but drops below $50\%$ in layers 0, 1. A similar observation has been reported for hybrid RoPE attention design [27]. This finding indicates that, even when RoPE is completely omitted, latent space tokens can still preserve almost all of the full attention scores across the majority of layers.

## 4 Sparse Attention in Latent Space Framework

### 4.1 SALS Framework

In this section, we introduce our proposed SALS, which leverages a low-rank projection matrix to transform multi-head pre-RoPE keys into a compact single-head latent space, thereby markedly shrinking the memory footprint. To avoid the prohibitive cost of reconstructing the entire key cache, SALS estimates attention scores directly in the latent space, selects the top-$N_c$ most critical tokens, and reconstructs only their corresponding keys. RoPE is only applied to the reconstructed keys, which are then used to compute the exact attention weights. Figure 3 provides a high-level summary of the SALS pipeline. Here, we use the top-k index $[2, 7]$ to indicate the selected tokens. The selected tokens are then reconstructed and reshaped to multi-head keys for sparse attention. The SALS framework is pipelined with three stages: KV cache compression to latent space, critical token selection in latent space and selective reconstruction from latent space for the later sparse attention.

### 4.2 Multi-head KV cache Compression to Latent Space

In the offline calibration process, we select a small calibration dataset from the pre-training corpus and collect its pre-RoPE key tensors. Let $\mathbf{K} \in \mathbb{R}^{s \times nd}$ denote the stacked multihead keys of the $s$ calibration sequences. We begin by computing the empirical covariance of the calibration keys: $\mathbf{C} = \mathbf{K}^\top \mathbf{K}$. Applying an eigenvalue decomposition, $\mathbf{C} = \mathbf{U}\mathbf{\Sigma}\mathbf{U}^\top$, and selecting the leading $r$ eigenvectors $\mathbf{U}_r$ yields the optimal rank-$r$ projection matrix for the latent space. Note that the key vectors $\mathbf{K} \in \mathbb{R}^{s \times nd}$ are reshaped by merging the head number with head dimension. We provide the rational using the following lemma.

**Lemma 1.** *For any column-orthonormal matrix $\mathbf{U} \in \mathbb{R}^{h \times r}$ satisfying $\mathbf{U}^\top \mathbf{U} = \mathbf{I}_r$, we define the captured variance by $\mathbf{U}$ as $E(\mathbf{U})$. For the multi–head joint projection, all possible projection matrix set is defined as $\mathcal{U}_r := \{\mathbf{U} \in \mathbb{R}^{nd \times r} \mid \mathbf{U}^\top \mathbf{U} = \mathbf{I}_r\}$, and $\mathcal{B}_r := \{\mathrm{diag}(\mathbf{U}_1, \ldots, \mathbf{U}_n) \mid \mathbf{U}_i \in$*

---

**Algorithm 1** Sparse Attention in Latent Space

---

**Input:** Pre-RoPE query $\mathbf{q} \in \mathbb{R}^{nd}$, key $\mathbf{k} \in \mathbb{R}^{nd}$, value $\mathbf{v} \in \mathbb{R}^{nd}$, Projection matrix $\mathbf{U}_r \in \mathbb{R}^{nd \times r}$,
    Low-rank key cache $\widetilde{\mathbf{K}} \in \mathbb{R}^{(S-1) \times r}$, quantised value cache $\widehat{\mathbf{V}} \in \mathbb{R}^{(S-1) \times nd}$, Sparsity budget $k$,
    Approx latent rank $r^\star$

**Output:** Attended output $\mathbf{y} \in \mathbb{R}^{nd}$

1: **Function** SPARSEATTENTION($\mathbf{q}, \mathbf{k}, \mathbf{v}, \mathbf{U}_r, \widetilde{\mathbf{K}}, \widehat{\mathbf{V}}, k$)
2:    $\tilde{\mathbf{q}} \leftarrow \mathbf{q}\mathbf{U}_r; \quad \tilde{\mathbf{k}} \leftarrow \mathbf{k}\mathbf{U}_r$               # Project new token to $r$-dim. latent space
3:    $\widetilde{\mathbf{K}} \leftarrow \mathrm{concat}(\widetilde{\mathbf{K}}, \tilde{\mathbf{k}}); \quad \widehat{\mathbf{V}} \leftarrow \mathrm{concat}(\widehat{\mathbf{V}}, \mathbf{v})$
4:    $\mathbf{p}' \leftarrow \tilde{\mathbf{q}}_{r^\star} \widetilde{\mathbf{K}}_{r^\star}^\top$                             # Cheap similarity on top-$r^\star$ dims
5:    $\mathcal{C} \leftarrow \mathrm{TopK}(\mathbf{p}', k)$                           # Selectr top–$k$ critical token index
6:    $\mathbf{K}_\mathcal{C} \leftarrow \widetilde{\mathbf{K}}_\mathcal{C} \mathbf{U}_r^\top$                         # Reconstruct critical latent key cache
7:    $\mathbf{q}^R \leftarrow \mathrm{RoPE}(\mathbf{q}); \quad \mathbf{K}_\mathcal{C}^R \leftarrow \mathrm{RoPE}(\mathbf{K}_\mathcal{C})$    # Standard flash attention (FA) computation
8:    $\mathbf{p} \leftarrow \mathrm{softmax}(\mathbf{q}^R \mathbf{K}_\mathcal{C}^{R^\top} / \sqrt{d})$
9:    $\mathbf{y} \leftarrow \mathbf{p}\,\widehat{\mathbf{V}}_\mathcal{C}$
10:   **return** $\mathbf{y}$
11: **End Function**

---

$\mathbb{R}^{d \times r'}$, $\mathbf{U}_i^\top \mathbf{U}_i = \mathbf{I}_{r'}\}$, *where* $r' = r/n$, *is all possible projection matrix set for per–head projection. For all situations, the optimal* $\mathbf{U}$ *from multi–head joint projection matrix can capture more enery then per–head projection matrix:*

$$\max_{\mathbf{U} \in \mathcal{U}_r} E(\mathbf{U}) \quad \geq \quad \max_{\mathbf{U} \in \mathcal{B}_r} E(\mathbf{U})$$

*Proof.* Pick any $\mathbf{U} = \mathrm{diag}(\mathbf{U}_1, \ldots, \mathbf{U}_n) \in \mathcal{B}_r$ with $\mathbf{U}_i^\top \mathbf{U}_i = \mathbf{I}_{r'}$. Because the blocks occupy disjoint coordinate ranges, their product is blockdiagonal:

$$\mathbf{U}^\top \mathbf{U} = \mathrm{diag}(\mathbf{U}_1^\top \mathbf{U}_1, \ldots, \mathbf{U}_n^\top \mathbf{U}_n) = \mathrm{diag}(\mathbf{I}_{r'}, \ldots, \mathbf{I}_{r'}) = \mathbf{I}_r.$$

Hence $\mathbf{U}$ is column-orthonormal and thus $\mathbf{U} \in \mathcal{U}_r$; consequently $\mathcal{B}_r \subseteq \mathcal{U}_r$. Since every candidate in $\mathcal{B}_r$ is feasible for the joint search $\mathcal{U}_r$, maximizing over the larger set cannot yield a smaller value. $\quad \square$

### 4.3 Critical Token Selection in Latent Space

As discussed in previous works [25, 29], tokenwise attention is inherently sparse. However, identifying the relevant (*critical*) tokens fast enough remains the main bottleneck of most sparse-attention schemes. Guided by the observation in Sec 3.2 that pre-RoPE latent keys already reveal the sparsity pattern of the middle layers, we forgo the expensive full-rank query-key multiplication and instead work in the latent space.

More specifically, let $\mathbf{U}_r \in \mathbb{R}^{nd \times r}$ be the joint projector from Equation 1 and let $r^\star \ll r$ for scoring. We project the query once, $\tilde{\mathbf{q}} = \mathbf{U}_r^\top \mathbf{q} \in \mathbb{R}^r$, and keep only its leading coordinates $\tilde{\mathbf{q}}_{:r^\star} \in \mathbb{R}^{r^\star}$. Each cached key already stores its full latent vector $\tilde{\mathbf{k}}_j = \mathbf{U}_r^\top \mathbf{k}_j$, from which we extract the first $r^\star$ dimensions, $\tilde{\mathbf{k}}_{j,:r^\star} \in \mathbb{R}^{r^\star}$. The approximate attention score of token in position $j$ is a cheap inner product

$$s_j \;=\; \tilde{\mathbf{q}}_{:r^\star}^\top \, \tilde{\mathbf{k}}_{j,:r^\star}$$

Thus, critical tokens are identified directly from the existing key cache, without extra storage and at a fraction of the original compute.

### 4.4 Selective Reconstruction from Latent Space for Sparse Attention

Sparse attention is defined by restricting the standard attention computation to a selected subset of keys and values. Given a query matrix $\mathbf{Q} \in \mathbb{R}^{n \times d}$ and a full set of key and value matrices of length $s$, we identify a subset of salient indices

$$\mathcal{C} = \{i_1, i_2, \ldots, i_{N_c}\} \subseteq \{1, \ldots, s\}, \quad N_c \ll s, \tag{4}$$

where $N_c$ denotes the number of selected positions on the token sequence. The corresponding sub-matrices $\mathbf{K}_\mathcal{C} \in \mathbb{R}^{N_c \times n \times d}$ and $\mathbf{V}_\mathcal{C} \in \mathbb{R}^{N_c \times n \times d}$ are extracted based on the selected indices $\mathcal{C}$. Here, the salient indices $\mathcal{C}$ are obtained via critical token selection in latent space. The sparse attention output is then computed by restricting the standard attention to the selected tokens:

$$\mathbf{p}_\mathcal{C} = \text{softmax}\left(\frac{\mathbf{Q}\mathbf{K}_\mathcal{C}^\top}{\sqrt{d}}\right), \qquad \mathbf{y} = \mathbf{p}_\mathcal{C}\mathbf{V}_\mathcal{C}. \tag{5}$$

This method works well since the softmax operation normalizes the input distribution so that the output sums to 1 and only a few positions (or tokens) in $\mathcal{C}$ dominate with large values. The rest scores after softmax closes to 0. The whole SALS algorithm is summarized in Algorithm 1.

### 4.5 Performance Discussion

As the roofline model [8] suggests, the performance of a GPU is bounded by either computation or memory bandwidth. In attention computation, performance is commonly bounded by memory bandwidth [7], which is primarily consumed by KV cache data movement. For a context of $s$ tokens and feature dimension $d$, full attention needs to transfer $2sd$ elements of keys and values. On the other hand, for the SALS framework, critical token selection and sparse attention both involve data movement. Critical token selection requires $r^\star \ll d$ latent dimensions, so the query–key dot products read $sr^\star$ elements. After selecting the top–$k$ tokens, SALS only needs to access low-rank key and value caches, each of size $kr$. The second pass therefore moves $2kr$ elements.

Combining this two phases, the sparse variant moves $sr^\star + 2kr$ scalars, so its memory-bound speed-up is: $\frac{2sd}{sr^\star+2kr} = \frac{1}{d_{r^\star}/2 + d_r\,k_s}$. Here $d_{r^\star} = r^\star/d$, $d_r = r/d$, $k_s = k/s$ denote the latent-approximated-score ratio, the low-rank ratio, and the token sparsity, respectively. We develop a `Triton` kernel that fuses token selection, reconstruction and RoPE rotation into a single pass. This fused reconstruct–RoPE kernel reduces memory traffic by $7.69\times$ to $14.28\times$, depending on the chosen sparsity level and low-rank compression ratio, compared to the standard FlashAttention implementation.

## 5 Experiment

### 5.1 Setup

**Models and tasks.** We evaluate our SALS framework on two main-stream 7B chat models: LLaMA2–7B–Chat [23], that employs multi–head attention (MHA), and Mistral–7B–v0.2 [11], that uses grouped–query attention (GQA). To assess accuracy, we report results on the reasoning benchmark GSM8K [6], the conversational benchmark CoQA [16], and the 16 English subsets of LongBench [2] that probe long–context understanding. All experiments are conducted on a machine with Xeon(R) Platinum 8336C CPU, one GPU (ampere architecture), and 128G RAM.

Table 2: Evaluation on GSM8K and CoQA datasets for LLaMA2–7B–chat

| Method | GSM8K (strict/flexible)↑ | CoQA↑ | Memory Access↓ | Comp. ratio↓ |
|---|---|---|---|---|
| baseline | 0.2335 / 0.2335 | 0.5997 | 1.00 | 1.00 |
| KIVI-4 | 0.2297 / 0.2305 | 0.5992 | 0.31 | 0.31 |
| KIVI-2 | 0.2047 / 0.2047 | 0.6010 | 0.19 | 0.19 |
| Palu-30%(3bit) | 0.1713/0.1759 | 0.5938 | 0.14 | 0.14 |
| Palu-50%(3bit) | 0.0614/0.0879 | 0.5560 | 0.09 | **0.09** |
| **SALS-25%** | **0.2312 / 0.2343** | 0.5975 | 0.13 | 0.28 |
| **SALS-12.5%** | 0.2176 / 0.2229 | **0.6070** | **0.07** | 0.15 |

**Baselines.** We compare SALS with several state of the art baseline in two directions. For KV–cache compression we select Palu (low–rank projection) [4] and KIVI (quantization) [14]. For token–sparse decoding we include Double Sparse [29], Hshare [25], and Loki [20]. For the KV–cache compression baselines, we evaluate Palu at every reported compression ratio, including its variants that apply the optional quantisation step. For KIVI we evaluate both official settings: 4 bit and 2 bit. For the token–sparse baselines, sparsity is introduced only during the decoding phase, and we enforce the same overall sparsity ratio as Hshare to ensure a fair comparison.

**Compression setup and calibration.** To obtain the latent projection matrix, we randomly sample 512 sequences of length 4096 from the C4 corpus [9] and compute it offline. We apply multi-head *joint* compression to the key cache at two ratios: $d_r = 25\%$ and $d_r = 12.5\%$. For latent scoring,

Table 3: Evaluation on LongBench datasets

| Model | Method | Single-QA | Multi-QA | Summari-zation | Few-shot | Synthetic | Code | Avg | Memory Access↓ |
|---|---|---|---|---|---|---|---|---|---|
| LLaMA2-7B-chat | baseline | 24.50 | 22.18 | 23.64 | 62.80 | 6.50 | 56.29 | 32.65 | 1.00 |
| | KIVI-4bit | **24.76** | 22.22 | **23.40** | 62.67 | 6.75 | **56.42** | **32.70** | 0.31 |
| | Palu-30% (3bit) | 22.36 | 22.01 | 22.66 | 60.39 | **7.50** | 50.53 | 30.91 | 0.13 |
| | SALS-25% | 24.37 | **22.50** | **23.40** | **63.01** | 6.50 | 53.80 | 32.26 | **0.11** |
| | KIVI-2bit | 23.11 | 21.40 | 22.67 | 62.81 | **5.75** | **55.84** | 31.93 | 0.19 |
| | Palu-50% (3bit) | 20.48 | 22.18 | 21.44 | 54.62 | **5.75** | 37.07 | 26.92 | 0.09 |
| | SALS-12.5% | **24.81** | **22.52** | **23.08** | **62.84** | 5.00 | 53.58 | **31.97** | **0.06** |
| Mistral-7B-v0.2 | baseline | 36.43 | 29.65 | 28.06 | 66.72 | 44.87 | 52.96 | 43.12 | 1.00 |
| | KIVI-4bit | 36.34 | 29.85 | **28.09** | **66.81** | **44.67** | **52.90** | **43.11** | 0.31 |
| | Palu-30% (3bit) | 29.48 | 36.40 | 27.20 | 65.73 | 53.19 | 40.77 | 34.74 | 0.13 |
| | SALS-25% | **36.61** | **29.92** | 27.57 | 66.63 | 43.54 | 52.49 | 42.79 | **0.11** |
| | KIVI-2bit | **35.34** | 28.63 | **27.72** | 66.77 | **39.68** | **52.63** | **41.80** | 0.19 |
| | Palu-50% (3bit) | 26.73 | **32.72** | 25.73 | 63.25 | 18.57 | 44.43 | 35.71 | 0.09 |
| | SALS-12.5% | 35.18 | 29.88 | 27.18 | **66.98** | 35.32 | 51.39 | 40.99 | **0.06** |

we simply set $r^\star = 0.5r$ for all models. Since value tensors are almost full rank and play a pivotal role in attention, we forgo low-rank projection for them and instead perform *channel-wise group quantisation* that mirrors the key-cache setting (4 bit at 25% and 2 bit at 12.5%).

Following KIVI, we adopt a mixed high–precision / low–precision scheme: tokens in the most recent window are compressed by only 50%, whereas all preceding tokens are compressed according to the target ratio of the experiment. The high–precision window is aligned with the sparsity window: when the sparse mechanism always selects the most recent $w$ tokens, the compression stage likewise keeps those same $w$ tokens in high precision. Based on the observations in Figure 2, we skip the sparsification for layers 0, 1, and 31 across all models to ensure more accurate compression and sparsity results.

## 5.2 KVcache Compression Comparison

**Implementation details.** For the dataset GSM8K and CoQA, we always keep the most recent $w = 128$ tokens and decode the remaining context at a fixed sparsity of $1/4$. For the dataset LongBench, LLaMA2–7B–Chat supports a 4 k context window, whereas Mistral–7B–v0.2 supports 32 k. To equalise the average sparsity at $1/8$ on this benchmark, we retain $N_c = 512$ tokens for LLaMA2–7B–Chat and $N_c = 1024$ tokens for Mistral–7B–v0.2. For LLaMA2–7B–Chat we follow the Hshare [25] configuration ($x = 16$ sink tokens, $y = 432$ critical tokens, $z = 64$ recent tokens) and simply double each count for Mistral–7B–v0.2.

**Results.** Tables 2 shows the evaluation on the benchmark (GSM8K, CoQA) on LLaMA2–7B–chat model. We can see that although Palu achieves the highest compression rate, the acurracy drop is non-negligible, especillay for the GSM8K dataset. SALS provides two setting results with 25% KV cache compression rate (SALS–25%) and 12.5% compression rate (SALS–12.5%). SALS–25% achieves the best accuracy compared to KIVI and Palu, with neligible loss compared to the baseline. Tables 3 compares task-level performance across six LongBench categories, alongside normalized memory access costs. The first three columns reflect core reasoning tasks, where SALS achieves consistently strong performance. Notably, SALS-12.5% retains competitive accuracy while reducing memory access to just 6% of the baseline. This indicates that SALS preserves informative tokens more effectively. The memory-access savings translate into practical latency improvements, as fewer KV memory lookups reduce attention computation overhead.

## 5.3 Token-Sparse Comparison

Since SALS incorporates a sparsification stage, we also compare it with state of the art sparse decoders, such as Double Sparse, HShare and Loki in LongBench datasets.

**Implementation details.** We use the same sparse setting as Sec 5.2 for all methods, which have $x = 16$ sink tokens, $y = 432$ critical tokens and $z = 64$ recent tokens, achieving sparsity on LLaMA2–7b–chat at $1/8$.

Table 4: Comparison of Token Sparse Methods on `LLaMA2-7B-chat` Using LongBench Tasks

| Method | Single-QA | Multi-QA | Summari-zation | Few-shot | Synthetic | Code | Avg | Memory Access↓ |
|---|---|---|---|---|---|---|---|---|
| baseline | 24.50 | 22.18 | 23.64 | 62.80 | 6.50 | 56.29 | 32.65 | 1.00 |
| Double Sparse | 24.78 | **22.72** | **24.70** | 61.84 | 4.17 | 51.66 | 31.64 | 0.16 |
| HShare | 24.59 | 22.18 | 24.54 | 61.69 | 4.74 | 53.26 | 31.83 | 0.14 |
| Loki | 24.57 | 18.09 | 24.37 | **63.43** | 4.75 | 56.49 | 31.95 | 0.19 |
| SALS-25% | 24.37 | 22.50 | 23.40 | 63.01 | **6.50** | **53.80** | **32.26** | 0.11 |
| SALS-12.5% | **24.81** | 22.52 | 23.08 | 62.84 | 5.00 | 53.58 | 31.97 | **0.06** |

Table 5: Performance comparison of baseline and SALS methods on the RULER dataset with Llama3.1-8B-Instruct (4k sequence length).

| Method | avg | S1 | S2 | MK1 | MK2 | MV | MQ | FEW | QA1 | QA2 |
|---|---|---|---|---|---|---|---|---|---|---|
| Baseline | 81.60 | 99.6 | 96.0 | 94.6 | 73.6 | 93.85 | 96.9 | 68.47 | 69.6 | 41.8 |
| SALS-25% | 80.81 | 99.6 | 95.2 | 94.2 | 65.8 | 93.2 | 96.4 | 71.53 | 70.2 | 41.2 |
| SALS-12.5% | 75.86 | 97.4 | 93.8 | 92.8 | 42.2 | 84.05 | 93.05 | 72.53 | 67.8 | 39.14 |

**Result.** Table 4 shows that, on LongBench, SALS with 25% key compression matches the average accuracy of the baseline while issuing only about 11% of the baselines memory traffic. When the compression is tightened to 12.5%, the Memory–Access Ratio drops to just 5.8%, yet SALS still outperforms all competing sparse methods in accuracy. These results suggest that the latent-space scoring mechanism selects critical tokens more accurately than existing heuristics, enabling KV-cache compression and token sparsity to coexist without loss of accuracy.

## 5.4 RULER Benchmark

**Dataset.** *RULER* [10] is a recently proposed long-context benchmark designed to evaluate the reasoning, retrieval, and compositional understanding capabilities of large language models across a wide range of input lengths and retrieval patterns. It decomposes the evaluation into fine-grained retrieval types, including *single-key*, *multi-key*, *multi-value*, and *multi-query* tasks, along with few-shot and question-answering (QA) subtasks. Compared with LongBench, RULER provides a more detailed breakdown of retrieval behaviors, making it well-suited for assessing token selection accuracy and KV-cache compression performance.

**Implementation details.** The experimental setup follows the same configuration as in Sec. 5.2. We evaluate all methods on a 128k-token context, applying an 8× sparsity ratio, i.e., selecting 16k active tokens during inference. The retained-to-pruned token ratio is identical to that used in the LongBench experiments, ensuring consistent compression levels and comparable evaluation metrics. All experiments are conducted on the **LLaMA 3.1–8B–Instruct** model with a 4k sequence length.

**Result.** Table 5 summarizes the performance of SALS on the RULER dataset. At a 25% compression ratio, SALS achieves nearly identical average accuracy to the baseline (80.81 vs. 81.60), demonstrating its high fidelity in token selection and KV-cache reconstruction. Performance remains consistent across most retrieval subtasks, including *NIAH-Single-1/2*, *Multi-Value*, and *Multi-Query*, indicating that latent-space sparsification preserves key contextual semantics even under substantial cache reduction.

When the compression is further tightened to 12.5%, accuracy degradation mainly occurs in *NIAH-Single-1* and *Multi-Key-2*, where the retrieval dependency is strongest. Nevertheless, the model still achieves competitive overall performance (75.86 average) and stable scores on *Few-shot* and *QA* tasks. These results suggest that the proposed latent-space attention mechanism can maintain accurate token importance estimation even under extreme sparsity, enabling effective KV-cache compression with minimal accuracy loss across retrieval-oriented benchmarks.

## 5.5 Efficiency Evaluation

**Inference details.** Following Hshare [25], all speed tests are performed on the PyTorch backend, with the Triton–based fused kernel as discussed in Sec 4.5. We evaluate two aspects: *(i)*

Table 6: Attention Operator Latency (ms) across Methods and Input Configurations

| Config (ms) | Flash-attn [7] | Loki [20] | Double-sparse [29] | Hshare [25] | SALS-25% | SALS-12.5% |
|---|---|---|---|---|---|---|
| bs=8, 1k | 0.230 | 0.248 | 0.138 | 0.134 | $0.409 \pm 0.093$ | $0.357 \pm 0.051$ |
| bs=8, 2k | 0.830 | 0.464 | 0.237 | 0.430 | $0.430 \pm 0.018$ | $0.359 \pm 0.012$ |
| bs=8, 4k | 1.630 | 1.102 | 0.724 | 0.576 | $0.530 \pm 0.008$ | $0.439 \pm 0.016$ |
| bs=16, 1k | 0.440 | 0.452 | 0.223 | 0.134 | $0.415 \pm 0.008$ | $0.360 \pm 0.011$ |
| bs=16, 2k | 1.630 | 0.864 | 0.434 | 0.246 | $0.552 \pm 0.040$ | $0.444 \pm 0.019$ |
| bs=16, 4k | 3.230 | 2.101 | 1.319 | 1.067 | $0.757 \pm 0.026$ | $0.565 \pm 0.013$ |

self–attention latency and *(ii)* end–to–end generation speed–up. Self–attention latency is compared against FlashAttention–v2 [7], whereas end–to–end throughput is benchmarked against GPT–Fast [13]. Experiments use batch sizes of 8 and 16 and sequence lengths of $1\,k$, $2\,k$, and $4\,k$ tokens; the sparsity ratio is fixed at $1/8$ for all methods.

**Result.** Tables 6 reports our attention latency comparisons. Mean and varaince are reported using 1000 repetions. Thanks to the added token sparsity, SALS accelerates the standalone attention operator by $7, 46\times$ speedup for batch size (bs) =8 and 4k sequence length (4k). Similar performance speed is also observed when batch size is 16. SALS will introduce some overhead for short sequences (e.g. 1k) but for longer sequences the speed-up is significantly better than the state-of-the-art sparse algorithm. As discussed in Section 4.5, reducing memory access will improve the attention performance.

Table 7 shows the end-to-end throughput speed-up of SALS comparing to the GPT-Fast results. Note that batch size is chosen 4 for sequence 64k to avoid GPU out of memory. Mean and varaince are reported using 50 repetions. We observe that when sequence is 8k, the speed-up of SALS-12.5% is $2.13\times$ although the sparsity ratio is 1/8. This is due to the overhead of reconstructing the selected token. When the sequence is set 32k, the speed-up of SALS-12.5% is increasing

Table 7: Long Context End-to-End Performance Throughput Comparison (token/second)

| Bsz | Seq (k) | GPT-Fast | SALS-25% | SALS-12.5% |
|---|---|---|---|---|
| 8 | 4 | 118 | $154.1 \pm 7.1$ | $163.5 \pm 5.1$ |
| 8 | 8 | 70.6 | $128.6 \pm 5.1$ | $150.1 \pm 3.3$ |
| 8 | 16 | 38.3 | $102.5 \pm 0.3$ | $122.7 \pm 7.7$ |
| 8 | 32 | 19.8 | $67.97 \pm 0.8$ | $89.47 \pm 1.4$ |
| 4 | 64 | 9.2 | $32.92 \pm 0.3$ | $42.96 \pm 0.9$ |

and reaches $4.57\times$. Coupled with the strong accuracy of SALS, these results confirm that combining latentspace KVcache compression with token sparsification offers an effective tradeoff between speed and accuracy.

## 6 Conclusion

This paper investigates the KV cache compression based on the low-rank characteristics in the hidden dimension. We further discuss the increasing variance of keys due to applying rotary position embedding and propose to use the pre-RoPE keys to select the critical tokens. We observe that after the key vectors are transformed into the latent space, they largely maintain their representation across most layers and thereby the salient tokens are selected with high accuracy. Based on these insights, we propose the Sparse Attention in Latent Space (SALS) framework. The SALS framwork achieves compressed KV cache to the latent space, selects the critical token in the latent space with significantly less computation load and performs sparse attention on the selected tokens. By reconstructing only a small subset of important tokens, it avoids the overhead of full KV cache reconstruction. Experimental results demonstrate that SALS achieves SOTA performance by maintaining competitive accuracy under different settings and achieving 6.4-fold KV cache compression and 5.7-fold speed-up in attention compared to FlashAttention2 on 4K sequences. For the end-to-end throughput performance, we achieve 1.4-fold and 4.5-fold improvement compared to GPT-fast on 4k and 32K sequences respectively.

## Acknowledgment

This work is supported by the National Nature Science Foundation of China (No. 62376020) and the Fundamental Research Funds for the Central Universities (No. 2025JBZY011).

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

# A  Key Dimensianlity Analysis with RoPE

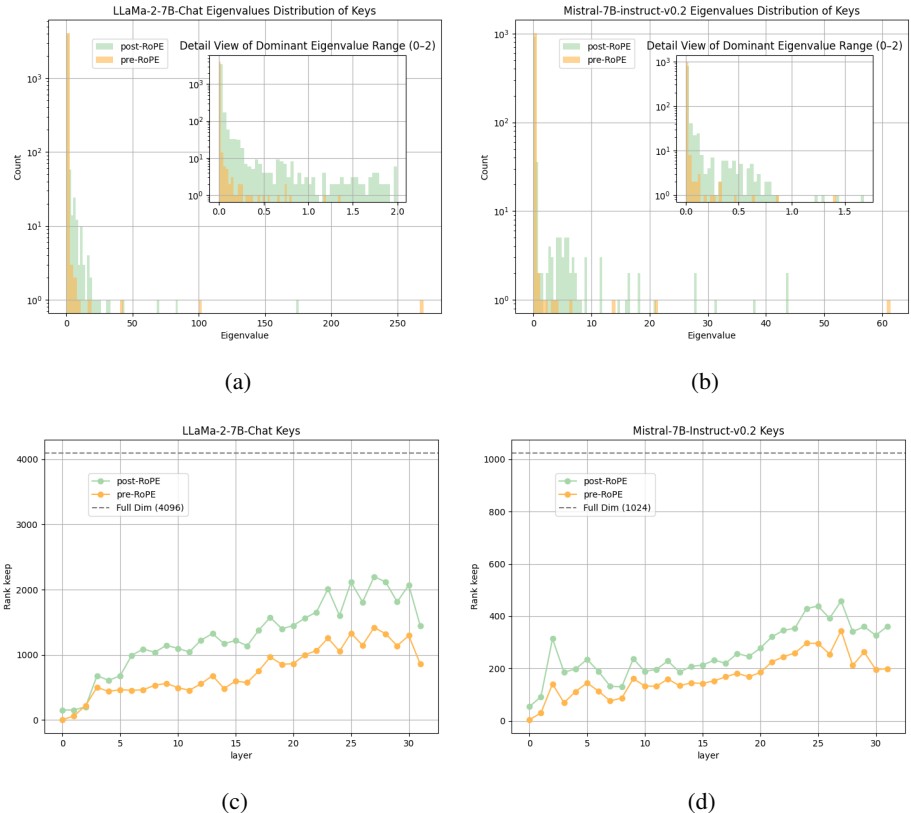

Figure 4: (a)–(b): Eigenvalue distributions of key covariance matrices in LLaMA-2-7B-Chat and Mistral-7B-Instruct-v0.2 before and after applying Rotary Position Embedding (RoPE). (c)–(d): Number of principal components required to retain 90% of the total variance across transformer layers, indicating changes in effective rank after RoPE.

In this section, we conduct a numerical analysis to quantify how rotary position embedding (RoPE) alters the principal-component structure of the key states. We perform principal component analysis (PCA) on the key tensors before and after applying RoPE, denoted as *pre-RoPE* and *post-RoPE*, respectively. Concretely, we first compute the covariance matrix of the key states and then carry out an eigenvalue decomposition. The magnitude of each eigenvalue reflects the contribution of its associated eigenvector: the presence of many small eigenvalues indicates that the corresponding representation is effectively low rank. To relate rank to the spectrum more systematically, we adopt the metrix introduced in Loki[20]:

$$\mathrm{Rank}_l(v) \ = \ \min\Big\{ d \in \mathbb{Z}_+ : \sum_{i=1}^{d} \lambda_l^{(i)} \ \geq \ \frac{v}{100}\Big\},$$

where $\lambda_l^{(i)}$ denotes the $i$-th largest eigenvalue of the covariance matrix for the keys in layer $l$. This definition specifies the smallest number of principal components needed to explain at least $v\%$ of the total variance, enabling a direct comparison of the intrinsic dimensionality before and after RoPE.

Figure 4(a)–(b) illustrate the eigen–value spectra of the first layer ($l = 0$) for Llama–2–7B–Chat and Mistral–7B–Instruct–v0.2 under the pre–RoPE and post–RoPE conditions. The pre–RoPE spectra exhibit a markedly larger number of small eigenvalues, whereas the post–RoPE spectra are shifted upward, corroborating our earlier finding that RoPE increases the overall variance. Figure 4(c)–(d) present the layer–wise $\mathrm{Rank}_l(90)$ values, i.e. the minimal dimensionality needed to retain $90\%$ of the variance. For both models, the post–RoPE condition consistently requires a higher rank, reflecting the broader spectra observed in panels (a)–(b). In addition, the required rank varies substantially across layers, indicating that a layer–adaptive rank selection scheme could further enhance compression efficiency.

