# OpenReview forum: "SALS: Sparse Attention in Latent Space for KV Cache Compression"
_NeurIPS.cc/2025/Conference — NeurIPS 2025 poster_

### Official Review · Reviewer_BamQ · 2025-06-27

**Clarity:** 3
**Significance:** 2
**Originality:** 2
**Rating:** 4
**Confidence:** 4

**Summary:**

The authors introduce Sparse Attention in Latent Space (SALS), a lightweight framework that reduces both storage and computation during inference by compressing the Key–Value (KV) cache. Rather than applying Rotary Position Embeddings (RoPE) before compression — which inflates the effective rank of key matrices — SALS first projects all multi-head key vectors into a shared, low-dimensional latent space via a rank-r eigen-projection. In this latent domain, approximate attention scores are computed using only the top coordinates, enabling selection of a small set of the most salient tokens. Only the corresponding top-k key–value pairs are then reconstructed back into their original space and attended using the standard mechanism, yielding a sparse attention over a highly compact cache representation. Empirical evaluation on LLaMA2-7B-Chat and Mistral-7B across benchmarks such as GSM8K, CoQA, and LongBench shows that SALS matches or surpasses the accuracy of FlashAttention2 and GPT-Fast baselines while achieving up to a 6.4× reduction in KV-cache size, a 5.7× speed-up in the attention operator at 4K context length, and overall 1.4–4.5× gains in end-to-end throughput for contexts ranging from 4K to 32K tokens. These results demonstrate that SALS can be seamlessly integrated into existing large-language-model pipelines to extend context lengths with minimal performance trade-offs.

**Questions:**

Q1: It would strengthen the empirical foundation to report SALS’s performance alongside contemporary sparse-attention methods, especially dynamic selection method like Quest or ArkVale.

Q2: Although you include a GQA-based model, the efficiency analysis may remain tailored to standard multi-head attention. Providing benchmarks on grouped-query attention layers — reporting memory access and throughput — would clarify SALS’s behavior under these widely used architectures.

Q3: Table 3 appears without narrative discussion, leaving its per-task metrics under-interpreted. A brief in-text analysis defining each column, highlighting key trends across Single-QA/Multi-QA/Summarization, and relating memory-access reductions to latency savings could improve clarity.

**Ethical Concerns:**

["NO or VERY MINOR ethics concerns only"]

**Final Justification:**

In the paper, the authors propose SALS that compresses the KV cache to reduce both storage and computation. The paper demonstrates promising speedup. The authors have addressed my concerns on the compatibility with GQA, and on more thorough comparison with Quest, Arkvale, etc, which is the main reason for me to raise the score to 4. However, the concern on limited novelty remains as SALS combines two well-explored ideas (latent compression and sparse token selection) without introducing fundamentally new theory.

**Limitations:**

Yes

**Quality:**

3

**Strengths And Weaknesses:**

Strengths:

+ The authors present a rigorously derived three-stage pipeline, including latent-space projection, critical-token selection, and selective reconstruction, that is mathematically sound and clearly justified. Their use of eigen-projection to compress pre-RoPE key vectors is underpinned by solid linear-algebraic reasoning, and the paper carefully measures the approximation error at each stage, demonstrating high fidelity to full attention.
+ By achieving up to 6.4× KV-cache compression and 1.4 - 4.5× end-to-end throughput gains on LLaMA2-7B-Chat and Mistral-7B, SALS addresses one of the most pressing bottlenecks in scaling transformer contexts.
+ The core idea of performing latent-space attention scoring prior to RoPE - thus sidestepping rank inflation - is a novel twist on existing compression and sparsity techniques. While latent projections and sparse attention have each seen prior study, their careful integration here constitutes a fresh contribution.

Weaknesses:

- Although the pipeline is robust, the overall methodological novelty is somewhat incremental: SALS combines two well-explored ideas (latent compression and sparse token selection) without introducing fundamentally new theory.

- The paper does not deeply investigate how SALS interacts with grouped-query attention (GQA) and multi-query attention (MQA) variants. Although a GQA-based model is included in the experimental suite, the efficiency analysis remains focused on the multi-head-attention (MHA) case and fails to articulate how the latent-space selection and sparse reconstruction behave under GQA/MQA. This omission leaves open the possibility that the reported performance gains may not generalize beyond the standard MHA setting.

- While throughput and memory gains are impressive, the experimental evaluation lacks comparisons to several leading sparse-attention frameworks, such as Quest, H2O and ArkVale that also target long contexts.

---

> ### Author Rebuttal · Authors · 2025-07-31
>
> > W1:  Although the pipeline is robust, the overall methodological novelty is somewhat incremental: SALS combines two well-explored ideas (latent compression and sparse token selection) without introducing fundamentally new theory.
> >
>
> **A:** Thank you for your comments.
>
> The latent compression is studied by many recent works such as Loki and Palu. The KV cache compression strategy and compression rate are not fully discussed, while the accuracy is hard to maintain.  To achieve higher accuracy yet better KV cache compression rate, our work proposes the joint-head compression comparing to the existing works of per-head compression (Loki) and group-head compression (Palu).  Joint-head has the advantage of preserving of the common principal components shared among different heads and thereby achieves better compression rate. Loki has not achieved any compression on KV cache whereas Palu achieves high compression rate (50%) but suffers from the accuracy loss (>5%).   On the other hand, our work achieves high compression rate (25%）by adopting joint-heads compression. Furthermore, with the sparse attention, the reconstruction overhead is significantly reduced with maintained accuracy (loss gap <1%).
>
> ---
>
> > W2: The paper does not deeply investigate how SALS interacts with grouped-query attention (GQA) and multi-query attention (MQA) variants. Although a GQA-based model is included in the experimental suite, the efficiency analysis remains focused on the multi-head-attention (MHA) case and fails to articulate how the latent-space selection and sparse reconstruction behave under GQA/MQA. This omission leaves open the possibility that the reported performance gains may not generalize beyond the standard MHA setting.
> >
>
> **A:** We are sorry for the confusion caused.  As shown in Figure 3, the token is selected based on the joint-head low-rank compressed latent space.  For the token selection stage, the latent keys and quires both only have one head. As such, there is only one head to select the token index from latent key cache and these indexes are later shared  by all the heads in the sparse attention.  Therefore, for GQA/MQA,  SALS works fine by sharing the critical tokens among the KV heads.
>
> Our experiment results in Table 5 are also performed based on the GQA setting (32 query heads and 8 KV heads).  We have shown clear speed-up (4.27X) comparing to the Flash-attention under the setting of batch size 16 and 16k sequence length.
>
> ---
>
> > Q1 It would strengthen the empirical foundation to report SALS’s performance alongside contemporary sparse-attention methods, especially dynamic selection methods like Quest or ArkVale.
> >
>
> > W3 While throughput and memory gains are impressive, the experimental evaluation lacks comparisons to several leading sparse-attention frameworks, such as Quest, H2O, and ArkVale that also target long contexts.
> >
>
> **A:** Thank you for highlighting recent sparse attention baselines. We have included experimental comparisons to **Quest** and **H2O**, which are leading dynamic selection methods. The results are summarized in **Table R1** below.
>
> **Table R1: Accuracy, latency, and memory access comparison with recent sparse attention baselines on LongBench.**
>
> | Method | Accuracy (avg / LongBench) | Latency (ms) | Memory Access ↓ |
> | --- | --- | --- | --- |
> | Flash‑Attn | 32.6519 | 3.23 | 1.00 |
> | H2O | 31.0919 | 0.85 | 0.12 |
> | Quest | 30.9719 | 2.21 | 0.25 |
> | SALS‑25% | 32.2628 | 0.757 | 0.11 |
> | SALS‑12.5% | 31.9706 | 0.565 | 0.06 |
>
> We compare SALS with FlashAttention, Quest, and H2O in terms of average accuracy (on LongBench), attention operator latency (ms), and normalized memory access. SALS achieves competitive accuracy with significantly lower memory cost and latency
>
> We also attempted to run **ArkVale**, but unfortunately its official implementation produced incorrect outputs without modification, and significant internal changes would be required to integrate it into our evaluation pipeline. Given the risk of introducing discrepancies, we chose not to include ArkVale results in this version.
>
> That said, SALS differs from ArkVale in several key aspects (e.g., latent-space selection, fully shared selection heads), and we plan to revisit this comparison in future work with a stabilized version of ArkVale.
>
> ---
>
> > Q2 Although you include a GQA-based model, the efficiency analysis may remain tailored to standard multi-head attention. Providing benchmarks on grouped-query attention layers — reporting memory access and throughput — would clarify SALS’s behavior under these widely used architectures.
> >
>
> **A:** Thank you for the question regarding SALS’s behavior under Grouped-Query Attention (GQA) and Multi-Query Attention (MQA).
>
> Our method is fully compatible with GQA/MQA architectures due to the **shared token selection mechanism**: token selection is performed in a joint latent space and is shared across all query heads. As a result, **the number of memory accesses does not increase with the number of query heads**, and **the selection cost remains constant regardless of the GQA/MQA configuration**.
>
> Furthermore, as discussed in Section 4.5 of our paper, the memory access formula is **independent of the number of attention heads**, ensuring that the efficiency gains of SALS naturally extend to GQA and MQA architectures.
>
> We provide additional runtime benchmarks in **Table R2** below, which reports the attention operator latency and normalized memory access cost (FlashAttention = 1.0) under a **GQA configuration** (32 query heads, 8 KV heads, batch size = 16, sequence length = 4k). These results are consistent with the end-to-end throughput evaluation presented in Table 6 of our main paper and further demonstrate that SALS achieves strong efficiency improvements under GQA settings.
>
> **Table R2: Latency and relative memory‑access ratios (Flash‑Attention = 1.0) at batch = 16, sequence length = 4 k**
>
> | **Method** | **Latency (ms)** | **Memory‑Access Ratio** |
> | --- | --- | --- |
> | Flash‑Attn | 3.230 | 1.00 |
> | Loki | 2.101 | 0.19 |
> | Double Sparse | 1.319 | 0.16 |
> | H‑Share | 1.067 | 0.14 |
> | SALS 25% | 0.757 | 0.11 |
> | SALS 12.5% | 0.565 | 0.06 |
>
> ---
>
> > Q3 Table 3 appears without narrative discussion, leaving its per-task metrics under-interpreted. A brief in-text analysis defining each column, highlighting key trends across Single-QA/Multi-QA/Summarization, and relating memory-access reductions to latency savings could improve clarity.
> >
>
> **A:** Thank you for the insightful suggestion. We agree that Table 3 would benefit from clearer narrative guidance. In the revised version, we will include a brief in-text analysis defining each column, highlighting trends across key task types (e.g., Single-QA, Multi-QA, Summarization), and explaining how reduced memory access contributes to improved efficiency. Specifically, we will add the following narrative:
>
> Table 3 compares task-level performance across six LongBench categories, alongside normalized memory access costs. The first three columns reflect core reasoning tasks, where SALS achieves consistently strong performance. Notably, SALS-12.5% retains competitive accuracy while reducing memory access to just 6% of the baseline. This indicates that SALS preserves informative tokens more effectively. The memory-access savings translate into practical latency improvements, as fewer KV memory lookups reduce attention computation overhead.
>
>
> To further support this, we will also include the Table R2 in the final version, which shows a clear correlation between memory-access reduction and latency savings across multiple methods:

---

### Official Review · Reviewer_v3oz · 2025-07-01

**Clarity:** 3
**Significance:** 3
**Originality:** 3
**Rating:** 4
**Confidence:** 4

**Summary:**

The paper introduces a framework named SALS to compress the KV cache and accelerate the generation of LLMs. It first analyzes the impact of RoPE on QK low rank and adopts a pre-RoPE low rank strategy. To build the framework, it transforms queries and keys into latent space, selects critical tokens in latent space and reconstructs them for sparse attention. Evaluations demonstrates the effectiveness and efficiency of SALS.

**Questions:**

See weaknesses.

**Ethical Concerns:**

["NO or VERY MINOR ethics concerns only"]

**Final Justification:**

The authors provide detailed clarifications about low rank value caches and a comparison with the NSA method. I have no further questions to the authors. Therefore, I will keep my rating.

**Limitations:**

See weaknesses.

**Quality:**

3

**Strengths And Weaknesses:**

Strengths:
1) The paper is clearly written and easy to follow, with sufficient empirical observations demonstrating the motivation for each design.

2) The proposed framework is carefully designed and demonstrates strong performance.


Weaknesses:
1) The value tensors remain in full rank. Despite the paper adopts quantization to compress value cache, it complicates the overall framework. Experimental results with low rank value caches should be provided.
2) A discussion between SALS and NSA [1] should be provided.
3) Incorporating comparisons of multiple value cache quantization methods (e.g. [2-3]) can further enhance the robustness of the SALS.


[1] Native Sparse Attention: Hardware-Aligned and Natively Trainable Sparse Attention. Arxiv 2502.11089.

[2] ZipCache: Accurate and Efficient KV Cache Quantization with Salient Token Identification. NeurIPS 2024.

[3] GEAR: An Efficient KV Cache Compression Recipe for Near-Lossless Generative Inference of LLM. Arxiv 2403.05527.

---

> ### Author Rebuttal · Authors · 2025-07-31
>
> > W1. The value tensors remain in full rank. Despite the paper adopts quantization to compress value cache, it complicates the overall framework. Experimental results with low rank value caches should be provided.
> >
>
> **A:** Thank you for your suggestions. We have added the value compression results in the Table R1 below. After the PCA compression, 50% and 25% values are stored. We find that the accuracy drop when performing value compression is larger than the quantization of SALS. The high sensitivity of value cache is also observed in [1].
>
> **Table R1: Performance on LongBench under different value compression settings.**
>
> | Method     | Single-Document QA | Single-Document QA | Summarization | Few-shot Learning | Synthetic | Code Completion | avg     |
> |------------|---------------------|---------------------|----------------|--------------------|-----------|------------------|----------|
> | baseline   | 24.50               | 22.18               | 23.64          | 62.80              | 6.50      | 56.29            | 32.65   |
> | SALS-25%   | 24.37               | 22.50               | 23.40          | 63.01              | 6.50      | 53.80            | **32.26** |
> | v-50%      | 21.64               | 22.06               | 22.32          | 61.59              | 6.00      | 49.07            | 30.45   |
> | v-25%      | 19.64               | 20.28               | 18.95          | 58.59              | 5.25      | 39.30            | 27.00   |
>
> [1] Yixin Ji, Yang Xiang, Juntao Li, Qingrong Xia, Zi Ye, Xinyu Duan, Zhefeng Wang, Kehai Chen, and Min Zhang. 2024. *Adaptive Feature-based Low-Rank Compression of Large Language Models via Bayesian Optimization.* In Findings of the Association for Computational Linguistics: EMNLP 2024, pages 4152–4168, Miami, Florida, USA. Association for Computational Linguistics.
>
> ---
>
> > W2. A discussion between SALS and NSA [1] should be provided.
> >
>
> **A:** Thank you for your suggestions. We will add the following discussion in our revised paper.
>
> Natively sparse attention (NSA) employs a dynamic hierarchical sparse strategy with coarse-grained token compression and fine-grained token selection. The major difference between NSA and SALS is that adopting NSA requires training from scratch, whereas SALS can be calibrated and achieved during the post-training phase. SALS shares the similar idea to perform the top-k token selection. However, NSA performs this token selection based on the trained compressed attention module. SALS performs the token selection based on the low-rank approximation.
>
> ---
>
> > W3. Incorporating comparisons of multiple value cache quantization methods (e.g. [2-3]) can further enhance the robustness of the SALS.
> >
>
> **A:** Thank you for your suggestions. We have added the ZipCache and GEAR quantization methods for comparisons using llama-7b-chat on the longbench dataset. As reported, GEAR, ZipCache and kivi quantization achieve similar accuracy under 4× (25%) compression rate, which illustrates the robustness of our SALS algorithms in incorporating different value cache quantization methods.
>
> **Table R2: Performance on LongBench under different value compression methods.**
>
> | Method | Single-Document QA | Single-Document QA | Summarization | Few-shot Learning | Synthetic | Code Completion | avg |
> | --- | --- | --- | --- | --- | --- | --- | --- |
> | baseline | 24.50 | 22.18 | 23.64 | 62.80 | 6.50 | 56.29 | 32.65 |
> | SALS-25%-kivi | 24.37 | 22.50 | 23.40 | 63.01 | 6.50 | 53.80 | 32.26 |
> | SALS-25%-gear | 23.15 | 22.56 | 22.85 | 63.63 | 6.25 | 55.22 | 32.28 |
> | SALS-25%-zipcache | 24.33 | 22.24 | 23.29 | 62.87 | 6.50 | 56.05 | **32.55** |

---

> > ### Comment · Reviewer_v3oz · 2025-08-05
> > **Response to the rebuttal**
> >
> > Thank you for the detailed clarifications. I have no further questions to the authors. Therefore I will keep my rating.

---

### Official Review · Reviewer_CDzv · 2025-07-02

**Clarity:** 2
**Significance:** 2
**Originality:** 2
**Rating:** 3
**Confidence:** 3

**Summary:**

The paper proposes SALS, a method to compress the KV cache in large language models by combining low-rank projection and token sparsity. Instead of reconstructing the full KV cache, SALS projects pre-RoPE keys into a compact latent space, selects top-k important tokens using lightweight scoring, and reconstructs only those tokens for final attention. This approach significantly reduces memory and compute without sacrificing accuracy. Experiments on Llama-2 and Mistral showcase that SALS achieve better accuracy and efficiency than existing baselines.

**Questions:**

- You project all multi-head keys into a shared latent space using a joint low-rank projection. Have you considered whether using per-head projections might improve token selection accuracy, despite the increased overhead?
- SALS is conceptually similar to Loki, which also performs low-rank key compression and token selection. Could you clarify the key differences in token scoring or reconstruction strategies between SALS and Loki?

**Ethical Concerns:**

["NO or VERY MINOR ethics concerns only"]

**Final Justification:**

I thank the authors for their detailed responses. Some of my concerns have been addressed and I have revised my score to 3. But I still have reservations about the paper's overall contribution. The technique of pre-RoPE compression has already been explored in prior works such as KIVI and KVQuant, and low-rank decomposition of the KV cache has been proposed in Loki. The proposed method appears to be a combination of existing techniques rather than a fundamental advancement in compute- and memory-efficient attention.

**Limitations:**

Yes.

**Paper Formatting Concerns:**

None.

**Quality:**

2

**Strengths And Weaknesses:**

### Strengths

1. This work provides good system support, showcasing great speedups relative to strong baselines such as FlashAttention.
2. The proposed approach is well-motivated through careful analysis.
3. Theoretical justifications are provided for using multi–head joint projection matrix.

### Weaknesses:

1. The paper reports memory access as a primary metric, but this does not directly capture the efficiency of the proposed method. It would be more informative to report latency or throughput for SALS and compare it with baseline methods.

2. Most experiments use relatively short context lengths (1K, 2K, 4K as noted on line 301). It would strengthen the evaluation to include results on longer contexts, such as 32K tokens or more, which are more representative of long-context use cases.

3. The contributions appear limited. The core idea of using low-rank projections to identify top-k tokens resembles existing methods like Loki [1]. The paper would benefit from a clearer explanation of what distinguishes SALS from these prior approaches.

[1] Loki: Low-rank Keys for Efficient Sparse Attention

---

> ### Author Rebuttal · Authors · 2025-07-31
>
> > W1 The paper reports memory access as a primary metric, but this does not directly capture the efficiency of the proposed method. It would be more informative to report latency or throughput for SALS and compare it with baseline methods.
> >
> **A:** Thank you for the valuable suggestions. Table R1 reports attention‑operator latency and the corresponding relative memory‑access ratios at **batch = 16, sequence length = 4 k**.
>
> As you pointed out, the latency of SALS does not reach its theoretical optimum for short sequences. Nevertheless, the measured latency mirrors the trend in the memory‑access ratio, and as the sequence length grows the fixed overhead becomes negligible; the latency ratio therefore converges toward the relative memory‑access ratio, as demonstrated in Table R2.
>
> **Table R1: Latency and relative memory‑access ratios (Flash‑Attention = 1.0) at batch = 16, sequence length = 4 k**
>
> | **Method** | **Latency (ms)** | **Memory‑Access Ratio** |
> | --- | --- | --- |
> | Flash‑Attn | 3.230 | 1.00 |
> | Loki | 2.101 | 0.19 |
> | Double Sparse | 1.319 | 0.16 |
> | H‑Share | 1.067 | 0.14 |
> | SALS 25% | 0.757 | 0.11 |
> | SALS 12.5% | 0.565 | 0.06 |
>
> ---
>
> > W2 Most experiments use relatively short context lengths (1K, 2K, 4K as noted on line 301). It would strengthen the evaluation to include results on longer contexts, such as 32K tokens or more, which are more representative of long-context use cases.
> >
>
> **A:** Thank you for pointing out the importance of long sequences. We have extended our experiments from 8 k up to **128 k tokens**; the new measurements are summarized in **Table R2** below (batch = 16, A100 80 GB, FP16).
>
> **Table R2: Attention operator latency and relative latency ratios of SALS‑25 % / SALS‑12.5 % versus Flash‑Attention at batch = 16, A100 80GB, FP16.**
>
> | **Seq Len (tokens)** | **Flash -attn(ms)** | **SALS‑25 % (ms)** | **SALS‑12.5 % (ms)** |
> | --- | --- | --- | --- |
> | 8 k | 6.072 | 1.247 | 0.843 |
> | 16 k | 11.850 | 2.282 | 1.469 |
> | 32 k | 23.922 | 4.212 | 2.600 |
> | 64 k | 51.056 | 8.294 | 5.082 |
> | 128 k | 102.628 | 16.240 | 9.830 |
>
> At a 4 k sequence length, SALS delivers a **5.7 ×** speed-up in attention operator latency. However, when the sequence length increases to 128 k, the fixed-overhead fraction becomes negligible, and the attention operator under **SALS-12.5%** achieves a **10.44 × acceleration** over FlashAttention.
>
> This speed-up aligns with the **theoretical latency reduction** discussed in Section 4.5 of our paper: since the dominant cost is memory access, the acceleration factor approximates the reciprocal of the memory-access ratio. For **SALS-12.5%**, the ratio is **0.06**, yielding a theoretical upper bound of **~16.6×**, which closely matches the trend observed in practice.
>
> ---
>
> > W3 The contributions appear limited. The core idea of using low-rank projections to identify top-k tokens resembles existing methods like Loki [1]. The paper would benefit from a clearer explanation of what distinguishes SALS from these prior approaches.
> >
>
> **A:** Thank you for your comments. Our work significantly differs from Loki in three manifolds:
>
> **1) Latent Space with RoPE**
>
> Loki obtains its low-rank projection matrix by performing PCA on either the pre-RoPE or post-RoPE keys. Regardless of how the projection is computed, Loki ultimately applies the transformation to **post-RoPE** keys and queries, and retains **full-rank** representations. This enables approximate attention score computation for token selection, but **does not perform actual compression**.
>
> As discussed in Section 3.1 of our paper, performing PCA after RoPE results in a significantly higher intrinsic rank, which leads to poor **low-rank approximation accuracy**. To address this, SALS applies low-rank projection **before** RoPE, enabling both accurate token selection and **key compression**. We also introduce a novel token selection mechanism in the pre-RoPE latent space, achieving better accuracy–latency trade-offs compared to Loki.
>
>
> **2) KV cache compression**
>
>  As mentioned in the limitation part of Loki paper, Loki has not achieved reduced memory usage of the KV cache. The KV cache is not compressed and stored in the low-rank fashion. Loki only used the low-rank queries and keys to perform the approximate attention. In the appendix E of Loki paper, Loki also proposed PCA-Attn, which directly used the reduced-dimensional scores. In this way, the KV cache is compressed but PCAAttn performs poorly comparing to all the baselines as summarized in Table 5 (Loki paper).
>
> In constrast, our work achieves KV cache **3.57× (1/0.28)** compression rate as shown in Table 2 (SALS paper). Moreover, because of the KV cache compression, the memory access is also reduced compared to Loki.
>
>
>
> **3) Joint-head compression vs. per-head compression**
>
> Loki's strategy is to perform per-head low-rank compression. As discussed in Fig. 2 from the Loki paper, the compression rate is around 50% for most models (e.g., for Llama2-7B, head dimension is compressed from 128 to 80 for most layers).
>
> Differently, our strategy is to perform joint-head compression, leading to more compression ratio. As shown in Table R2 of our paper, we can achieve **4× and 8× compression** while maintaining the accuracy.
>
> **Summary:** To achieve KV cache compression using the low-rank strategy brings new challenges to the existing method. Our work tackles this challenge by merging heads for compression and carefully integrating rotary embedding in the SALS architecture. We achieve **6.4× KV cache compression** while maintaining the accuracy on 4K sequences compared to FlashAttention 2.
>
> Reference: Singhania, Prajwal, et al. *"Loki: Low-rank keys for efficient sparse attention."* Advances in Neural Information Processing Systems 37 (2024): 16692-16723. https://arxiv.org/pdf/2406.02542
>
> ---
>
> > Q1: You project all multi-head keys into a shared latent space using a joint low-rank projection. Have you considered whether using per-head projections might improve token selection accuracy, despite the increased overhead?
> >
>
> **A:** Thank you for the insightful question. We explored this alternative and implemented a **per‑head** variant where each attention head is equipped with its own rank‑*r* projection matrix. To ensure a fair comparison, we kept the total projection rank constant: for *h* heads the per‑head rank is set to *r / h*, so both variants introduce the same number of projected dimensions. To ensure robustness, our results are averaged over 500 input samples and cover all attention layers.
>
> **Table R3:  Token‑selection alignment of joint vs per‑head low‑rank projections**
>
> | Metric ↑ | Joint (mean) | Per‑head (mean) |
> | --- | --- | --- |
> | **Overlap Score** | **0.931** | 0.812 |
> | **Overlap Rate** | **0.624** | 0.469 |
>
> To evaluate the token selection module, we report two metrics in Table R3: **Overlap Score**, which reflects the **effectiveness** of selection in capturing high-attention tokens (i.e., whether the selected tokens are valuable under full attention), and **Overlap Rate**, which measures the **accuracy** of selection by comparing against the top-*k* tokens in the full attention.
>
> As shown in Table R3, joint-head projections outperform per-head projections. While joint projections may sacrifice per-head specificity, they better preserve the globally important directions in the low-rank space, as discussed in Lemma 1. This results in higher overall token selection accuracy.
>
> In addition, as also observed in prior work (Palu), **joint-head projections have the advantage of preserving the common principal components shared across heads**, thereby achieving better compression efficiency under a constrained projection budget.
>
> ---
>
> > Q2: SALS is conceptually similar to Loki, which also performs low-rank key compression and token selection. Could you clarify the key differences in token scoring or reconstruction strategies between SALS and Loki?
> >
>
> **A:** Thank you for your comments. Besides the difference in the Latent space with RoPE,  KV cache compression, and joint-head compression mentioned in the previous reply, the token scoring and reconstruction strategies are also different:
>
> (1) Since Loki performs the compression on each head, the token scores are for each head and tokens are selected differently for each head. For our case, we perform the joint-head compression, where each head selects the same tokens.
>
> (2) For the reconstruction strategies, Loki performs no KV cache compression, and thereby it can directly select the chosen KV cache. However, for our SALS, the selected token is stored in the compressed fashion and only reconstructed when it is selected.

---

> ### Author Response · Authors · 2025-08-06
>
> Dear Reviewer CDzv,
>
> I would like to sincerely thank you for the careful review, the valuable comments and constructive suggestions, which help to improve the quality of our paper. I have addressed your concerns in the response above, including: (1) adding latency evaluations and extending experiments to longer sequences to better demonstrate the efficiency of our method; (2) clarifying our rationale for using joint-head low-rank projections and demonstrating their advantage over per-head projections; and (3) clarifying the key differences between our approach and prior work such as Loki in terms of latent space with RoPE, KV cache compression, and token selection strategy.
>
> Please feel free to let me know if you have any further questions or if anything remains unclear. I’d be happy to provide further clarification if needed.
>
> Best regards,
>
> the authors

---

> ### Comment · Area_Chair_3nrL · 2025-08-06
> **Engage in Discussion**
>
> Dear Reviewer CDzv,
> Thank you for reviewing this manuscript.
> Could you engage in the author-reviewer discussion after reading the author's response?

---

> ### Author Response · Authors · 2025-08-08
>
> Dear Reviewer CDzv,
>
> Thank you again for your review and for submitting the Mandatory Acknowledgement. I noticed that there had been no public reply following the acknowledgement, and I just wanted to check whether this might be due to a technical issue on OpenReview or some display limitation on our end. If there was a response that didn’t show up on our side, please kindly let us know.
>
> As the discussion period is coming to a close, we remain available to provide further clarification on any aspects that may still be unclear. We sincerely value your feedback and appreciate the opportunity to engage in this discussion.
>
> Best regards,
>
> the authors

---

> > ### Comment · Reviewer_CDzv · 2025-08-08
> >
> > Thank you for the additional comments. I have added comments in Final Justification but it did not show up. Sorry for the confusion. My comments regarding the rebuttal:
> >
> > I thank the authors for their detailed responses. Some of my concerns have been addressed and I have revised my score to 3. But I still have reservations about the paper's overall contribution. The technique of pre-RoPE compression has already been explored in prior works such as KIVI and KVQuant, and low-rank decomposition of the KV cache has been proposed in Loki. The proposed method appears to be a combination of existing techniques rather than a fundamental advancement in compute- and memory-efficient attention.

---

> > > ### Author Response · Authors · 2025-08-09
> > >
> > > Thank you for raising the connections to prior works. To clarify and avoid potential misunderstandings, we would like to highlight several distinctions:
> > >
> > > **(A) KIVI and KVQuant**
> > >
> > > - **KIVI** is a quantization-based compression approach and does not explore either pre-RoPE or post-RoPE applications. It therefore does not address the effect of RoPE placement on the quality of compressed representations.
> > > - **KVQuant** examines quantization both before and after RoPE, but it remains fundamentally different from low-rank methods. Quantization reduces numeric precision, whereas low-rank compression exploits redundancy in the latent subspace. These techniques operate on different assumptions and exhibit distinct accuracy–compression trade-offs, thus the conclusions from quantization studies cannot be assumed to hold for low-rank compression. Moreover, quantization and low-rank compression are orthogonal techniques, and can be combined — e.g., applying quantization to the key cache in the latent space — to further improve compression ratios.
> > >
> > > **(B) Loki**
> > >
> > > **B1. Pipeline differences**
> > > Due to the fundamentally different handling of RoPE, SALS diverges from Loki across all four key stages in the pipeline. These differences — in the selection of transforming the KV cache before or after RoPE, the dimensionality of stored keys, the basis for token selection, and the reconstruction strategy before attention — enable SALS to compress the key cache while preserving accuracy, resulting in both higher efficiency and better accuracy compared to Loki.
> > >
> > > | Step         | Loki | SALS | Key Difference |
> > > |--------------|------|------|----------------|
> > > | 1. Low-rank transformation | $\begin{aligned} q^R,k^R &= \mathrm{apply\\_rope}(q, k) \\\\ \hat{q}^R &= q^R P,\ \hat{k}^R = k^R P \end{aligned}$ | $\hat{q}_r = q P\_r,\ \hat{k}_r = k P\_r$ | SALS applies the low-rank transformation to $k$ **before** RoPE to avoid rank distortion introduced by positional encoding, whereas Loki applies it **after** RoPE using PCA transformation. |
> > > | 2. Key cache storage | $\text{Key cache} \mathrel{+}= \hat{k}^R$ | $\text{Key cache} \mathrel{+}= \hat{k}_r$ | Loki stores the PCA-transformed $\hat{k}^R$ **without rank reduction**, while SALS stores the low-rank $ \hat{k}\_r$ already **compressed to $r$ dimensions**, leading to lower memory cost. |
> > > | 3. Token selection | $\mathrm{Topk\\_token} = \mathrm{topk}(\\hat{q}^R_r \ \\hat{k}^R_r{}^{\\top})$ | $\mathrm{Topk\\_token} = \mathrm{topk}(\\hat{q}_r \ \\hat{k}_r{}^{\\top})$ | SALS performs token selection using the **pre-RoPE** low-rank $k$, whereas Loki uses the **post-RoPE** low-rank $k$. |
> > > | 4. Attention computation | $\mathrm{attn\\_scores} = \\hat{q}^R \ \\hat{k}^R\_{\\mathrm{sparse}}$ | $\begin{aligned} \mathrm{reconstruct\\_k} &= \\hat{k}^r\_{\\mathrm{sparse}} P_r^{\\top} \\\\  q^R, k\_{\\mathrm{sparse}}^R &= \mathrm{apply\\_rope}(q, \mathrm{reconstruct\\_k}) \\\\  \mathrm{attn\\_scores} &= q^R \ k\_{\\mathrm{sparse}}^R \end{aligned}$ | Loki directly computes attention between the transformed $q$ and sparse $k$, while SALS first reconstructs the sparse $k$, applies RoPE, and then computes attention. |
> > >
> > > **B2. PCA-Attn and empirical differences**
> > >    In Appendix E, Loki explores PCA-Attn, which extends its pipeline to compress the KV cache. However, PCA-Attn’s perplexity increases sharply compared to both the baseline and SALS:
> > >
> > >    **Table 1 – Perplexity (PPL) comparison at 25% compression**
> > >
> > >    | Baseline PPL | PCA-Attn PPL | SALS PPL |
> > >    |--------------|--------------|----------|
> > >    | 5.1102           | 243.2631         | 5.2141       |
> > >
> > >    Furthermore, Table 2 shows LongBench results, where even without KV cache compression, Loki’s accuracy is lower than SALS (which combines compression with sparsity):
> > >
> > >    **Table 2 – LongBench performance comparison**
> > >
> > >    | Baseline Score | Loki Score | SALS Score |
> > >    |----------------|------------|------------|
> > >    | 32.65 | 31.95 | 32.26 |
> > >
> > > These points demonstrate that SALS is not a trivial combination of prior techniques but instead introduces a **fundamentally different RoPE handling strategy** that prior methods (including **KIVI**, **KVQuant**, and **Loki**) do not address.
> > >
> > > By applying **low-rank compression before RoPE** and **optimizing the entire pipeline**, SALS achieves **substantial KV cache reduction without degrading perplexity**, even under aggressive compression ratios.
> > >
> > > Moreover, across both perplexity and long-context benchmarks, SALS consistently **outperforms Loki** (and its PCA-Attn variant), demonstrating that our approach delivers **higher efficiency and better accuracy** in realistic long-context scenarios.
> > >
> > > These findings highlight that SALS achieves efficient KV cache compression while maintaining **strong performance**, particularly in long-context scenarios.

---

### Official Review · Reviewer_YyV8 · 2025-07-04

**Clarity:** 3
**Significance:** 2
**Originality:** 3
**Rating:** 4
**Confidence:** 5

**Summary:**

SALS is a framework designed to improve inference efficiency for Large Language Models (LLMs) with long contexts by compressing the large Key-Value (KV) cache. It leverages two insights: RoPE increases key vector variance (raising rank), and key representations remain stable across layers. SALS projects the KV cache into a compact latent space using low-rank projection and performs sparse token selection via RoPE-free interactions, reconstructing only key tokens. This avoids full KV reconstruction overhead. Evaluated on LLaMA2-7b-chat and Mistral-7b, SALS achieves up to 6.4× KV compression, 5.7× faster attention, and up to 4.5× end-to-end speedup, while maintaining competitive accuracy.

**Questions:**

1. Can you provide numbers to justify Table 1?
2. Additional experiments as stated in the weaknesses.
3. InfiniGen, ShadowKV also employs low-rank compression for KV cache. Could you provide any insights about the core differences?.

**Ethical Concerns:**

["NO or VERY MINOR ethics concerns only"]

**Final Justification:**

This paper explores an interesting method for KV sparsity+low rank, which is promising. However, I feel it is similar to other KV low rank algorithms, although this work shows some advantages. In addition, this paper does not thoroughly compare with competitive baselines, such as Quest and ShadowKV, in benchmarks like RULER in a longer context (128K).

**Limitations:**

There are many limitations generally for sparse attention. For example, the speed up is limited by how much memory taken by KV cache. I think this paper does not discuss tlimitations in the work.

**Paper Formatting Concerns:**

No.

**Quality:**

2

**Strengths And Weaknesses:**

Strengths:
1. The paper is easy to follow.
2. The baseline implementations (GPT-Fast) are valid.
3. This paper discusses an important problem of low rank KV compression suffering severe accuracy degradation.

Weaknesses:

1. The evaluation models (Llama2-7b and mistral) are even not long-context models (Llama3.1 is more prefered).
2. The evaluation dataset is not standard. RULER, InfiniBench, and GSM-Infinite are more convincing.

---

> ### Author Rebuttal · Authors · 2025-07-31
>
> > Q1 Can you provide numbers to justify Table 1?
> >
>
> **A:** Thank you for you suggestions. We have provide the KV data movment, memory size and computation complexity comparision equation. These equations are used to support the claim low/median/high marks.
>
>
> **Table R1: KV data movement, memory cost and complexity comparison of quantization, low rank and token sparse methods with fixed, dynamic and local token selection strategies. We use $b$, $n$, $n_q$, $s$, $d$ to represent the batch size, KV head number, Q head number, sequence length and head dimension, respectively. We further use $s'$ and $s_{fixed}$ for top-k selected token length and fixed pattern length, respectively, for sparse attention. $r$ represents the compressed rank of KV cache.**
>
> | **Name** | **Methods** | **KV data movement** | **Memory size** | **Computation Complexity** | **Accuracy (gap)** |
> | --- | --- | --- | --- | --- | --- |
> | Palu | Low Rank | Median-$O(bsnr)$ | Low-$O(bsnr)$ | High-$O(bsnrd)+O(bns^2d)$ | Low--(1%-5%) |
> | Loki | Dynamic + Low Rank | Low-$O(bs'nd)$ | Median-$O(bsnd)$ | Median-$O(bns'^2d)$ | High--(0%-1%) |
> | StreamingLLM | Fixed pattern | Low-$O(bs_{fixed}nd)$ | Median-$O(bsnd)$ | Low-$O(bns_{fixed}^2d)$ | Low--(3%-5%) |
> | Quest | Dynamic | High-$O(bs'n_qd)$ | Median-$O(bsnd)$ | Median-$O(bns'^2d)$ | Median--(1%-2%) |
> | DS | Dynamic | Low-$O(bs'nd)$ | Median-$O(bsnd)$ | Median-$O(bns'^2d)$ | High--(0%-1%) |
> | Hshare | Dynamic | Low-$O(bs'nd)$ | Median-$O(bsnd)$ | Median-$O(bns'^2d)$ | High--(0%-1%) |
> | **SALS (Ours)** | Dynamic + Low Rank | Low-$O(bs'nr)$ | Low-$O(bsnr)$ | Median-$O(bns'^2d)$ | High--(0%-1%) |
>
> The accuracy gap is difficult to compared. For the Palu' accuracy, we take the experiments from Table 3 in our paper. StreammingLLM is based on Tabel 3 in Hshare [1] paper on the longbench dataset. For the rest of methods , the accuracy gap is taken from Table 4 in our paper.
>
> [1] Wu, Huaijin, et al. "HShare: Fast LLM decoding by hierarchical key-value sharing." The Thirteenth International Conference on Learning Representations. 2025.
>
> ---
>
> >
> > Q2 Additional experiments as stated in the weaknesses.
> >
>
> > W1 The evaluation models (Llama2-7b and mistral) are even not long-context models (Llama3.1 is more preferred).
> >
>
> > W2 The evaluation dataset is not standard. RULER, InfiniBench, and GSM-Infinite are more convincing.
> >
>
> **A:** Thank you for your suggestions. We have added benchmark results on the **RULER** dataset using **LLaMA3.1-8B-Instruct**.  **Table R2-a** and **Table R2-b** present our main results on 4k and 32k sequence lengths. Following the LongBench setting, we use an overall sparsity of 1/8. The results demonstrate that SALS retains high performance under long-context conditions, confirming its robustness.
>
> Due to the multimodal and video-based nature of **InfiniBench** and the lack of standardized implementations for **GSM-Infinite**, we focused our benchmark extension on **RULER**, which is fully text-based and compatible with our current evaluation pipeline. We plan to include GSM-Infinite and other long-context datasets in future evaluations as implementations mature.
>
> We selected **LLaMA3.1-8B-Instruct** to better reflect practical long-context use. We report results at a sequence length of **32k**, as even optimized attention implementations (e.g., FlashAttention) face memory constraints beyond this point on single-GPU setups. The 32k setting thus offers a meaningful and representative context length for evaluation.
>
> We will include these extended results in the revised version of the paper.
>
>
> **Table R2-a: Performance comparison of baseline and SALS methods on the RULER dataset with Llama3.1-8B-Instruct (4k sequence length).**
>
> | Method        | avg        | niah_single_1 | niah_single_2 | niah_single_3 | niah_multikey_1 | niah_multikey_2 | niah_multikey_3 | niah_multivalue | niah_multiquery | fwe    | qa_1  | qa_2 |
> |---------------|------------|----------------|----------------|----------------|------------------|------------------|------------------|------------------|------------------|--------|-------|------|
> | baseline | 92.8845    | 100            | 99.8           | 100            | 100              | 99.8             | 99.8             | 99.9             | 99.5             | 85.73  | 82.2  | 55   |
> | sals-25%      | 92.1045    | 100            | 99.6           | 99.8           | 100              | 100              | 99.8             | 99.7             | 99.45            | 78.8   | 81.2  | 54.8 |
> | sals-12.5%    | 90.6109    | 99.6           | 95.6           | 97.6           | 97.4             | 99.2             | 97.0             | 98.65            | 98.2             | 77.27  | 80.6  | 55.6 |
>
> **Table R2-b: Performance comparison of baseline and SALS methods on the RULER dataset with Llama3.1-8B-Instruct (32k sequence length).**
>
> | Method        | avg        | niah_single_1 | niah_single_2 | niah_single_3 | niah_multikey_1 | niah_multikey_2 | niah_multikey_3 | niah_multivalue | niah_multiquery | fwe    | qa_1  | qa_2 |
> |---------------|------------|----------------|----------------|----------------|------------------|------------------|------------------|------------------|------------------|--------|-------|------|
> | baseline_fp16 | 89.5618    | 91             | 98             | 100            | 98.8             | 98.4             | 98.8             | 98.75            | 98.1             | 80.33  | 76.4  | 46.6 |
> | sals-25%      | 89.2482    | 91             | 97.8           | 100            | 98.6             | 98.0             | 98.4             | 98.35            | 97.85            | 80.13  | 75.6  | 46.0 |
> | sals-12.5%    | 88.7755    | 93.4           | 97.6           | 99.2           | 98.2             | 97.2             | 93.0             | 95.3             | 98.3             | 83.13  | 75.2  | 46.0 |
>
> ---
>
> > Q3 InfiniGen, ShadowKV also employs low-rank compression for KV cache. Could you provide any insights about the core differences?
> >
>
> **A:** Thanks for your comments. Although our SALS share some similarities with InfiniGen and ShadowKV in employing low-rank compression for KV cache, there exist several core differences:
>
> **(1) The difference from InfiniGen.**  InfiniGen is built based on two observations: the attention input similarity and the skewed partial weights. Skewed partial weights is the most related to our work, however, skewed weight is designed to identify the large-value channels (outliers) in the queries and keys without considering the RoPE position embedding. As discussed in our work, RoPE leads to increasing rank. In the InfiniGen work, the skewing (low-rank) matrix is fused with weights, and the outliers in the queries and keys are element-wise added to select the top-30% channels for the approximated attention scores computations.
>
> In our work, we directly adopt the low rank keys and queries to compute the approximated attention scores. We do not fuse the orthogonal matrix into weights as discussed in our work, the RoPE operation introduces new variances and increases rank. Another key difference is that we compress the keys and only store the low-rank key cache, whereas InfiniGen stores the full rank keys in the CPU memory and performs prefetching of keys to avoid long communication latency.
>
> **(2) The differences fromShadowKV.** our method differs in two key aspects: the usage of low-rank compression and the strategy for critical token selection.
>
> a) shadowKV performs SVD-based low-rank decomposition for each token during decoding, producing token-specific low-rank matrices (A, B). While this may offer finer-grained reconstruction, it introduces additional online compression overhead and increases memory usage. In contrast, our method applies **offline PCA-based calibration** to derive a **shared low-rank projection matrix** across all tokens, which significantly reduces runtime cost and simplifies the memory layout during inference.
>
> b) ShadowKV selects a chunk of tokens (e.g. 8 tokens) and adopts chunk-level approximation (mean of each trunk) for selecting important tokens. However, such approximation is not accurate enough, therefore, a small set of outlier chunks also has to be stored. In our case, we perform token level approximation by utilizing a smaller rank than stored KV cache. Our selection is much simpler without the need to perform complex outlier selection and also achieve comparable accuracy.
>
> The discussion above will be added in our revised manuscript.
>
> ---
>
> > **L1** There are many limitations generally for sparse attention. For example, the speed up is limited by how much memory taken by KV cache. I think this paper does not discuss limitations in the work.
> >
>
> **A:** Thanks for your comments.  We will add the following discussion about the limitations in the revised manuscript.
>
> In this paper, we introduce the Sparse Attention in Latent Space (SALS) framework, which projects the KV cache into a compact latent space via low-rank projection and performs sparse token selection in this space for sparse attention. The benefits of adopting sparse attention is greatly depending on the sequence length. As shown in Table 5, when the sequence length is not long enough such as 1k, the sparse attention (latency 0.409ms) performs worse than the standard flash attention (latency 0.23ms). On the other hand, when the sequence length is 4k, sparse attention achieves 3.08X speed-up.
>
> The current paper has not explored the trade-off for the KV cache compression rate and the accuracy of token selection.  With consideration of the sequence length, there will be a sweet point to improve the compression rate without losing the accuracy.  Moreover,  the current paper considers a fixed compression rate and sparse ratio for all the layers. In the future work, we are interested to explore memory-performance-accuracy trade-offs on different layers.

---

> > ### Comment · Reviewer_YyV8 · 2025-08-04
> >
> > Thanks for the response. The authors need to (1) justify the difference with ShadowKV, Loki, and (2) report benchmarks for RULER (any chance for 128K and longer?). Besides, the authors address most of my concerns in the rebuttal. I will update my score to 4.

---

> > > ### Author Response · Authors · 2025-08-08
> > >
> > > Thank you for your response and for the helpful suggestions.
> > >
> > > While all three methods—our approach, ShadowKV, and Loki—leverage the low-rank structure of attention keys, our method differs fundamentally in both compression strategy and design choices.
> > >
> > > - **Compared to ShadowKV**, which performs token-wise online SVD over the entire prefill key matrix, our method uses offline PCA to compute a shared low-rank projection matrix. The token-specific SVD bases used in ShadowKV cannot be reused during decoding, as each new key would require re-computing the decomposition—an impractical cost. As a result, ShadowKV compresses only the prefill keys and stores all subsequent keys in full precision, leading to diminishing benefits during long decoding (e.g., chain-of-thought reasoning). In contrast, our shared PCA-based projector can be efficiently applied to all keys, including those generated during decoding, enabling consistent compression throughout the entire sequence.
> > >
> > > - **Compared to Loki**, which applies PCA in the post-RoPE space where the intrinsic rank is significantly higher, SALS performs compression and token selection in a more compressible pre-RoPE latent space. This design allows for more accurate low-rank approximation and enables actual KV cache compression. Loki retains full-rank key/value representations and does not compress the KV cache, as aggressive dimensionality reduction in the post-RoPE space leads to poor approximation. In contrast, SALS achieves 4× to 8× KV cache compression via joint-head low-rank projection, reducing both memory footprint and memory access costs during decoding, as shown in Table 2 of our paper.
> > >
> > > As a follow-up to your suggestion, we additionally report performance on **RULER with 128K context length** below:
> > >
> > > | Method      | avg     | niah_single_1 | niah_single_2 | niah_multikey_1 | niah_multikey_2 | niah_multivalue | niah_multiquery | fwe   | qa_1  | qa_2  |
> > > |-------------|---------|----------------|----------------|------------------|------------------|------------------|-------------------|--------|--------|--------|
> > > | baseline    | 81.60   | 99.6           | 96.0           | 94.6             | 73.6             | 93.85            | 96.9              | 68.47  | 69.6   | 41.8   |
> > > | SALS-25%    | 80.81   | 99.6           | 95.2           | 94.2             | 65.8             | 93.2             | 96.4              | 71.53  | 70.2   | 41.2   |
> > > | SALS-12.5%  | 75.86   | 97.4           | 93.8           | 92.8             | 42.2             | 84.05            | 93.05             | 72.53  | 67.8   | 39.14  |
> > >
> > > These results show that SALS maintains strong performance at 128K.
> > >
> > > We will include both the clarification on differences and the 128K RULER results in the final version of the paper.

---

### Author Response · Authors · 2025-08-09

Dear ACs, PCs, and all reviewers,

We would like to express our sincere gratitude to all the reviewers for their valuable comments and constructive feedback on our work.

Our work aims to reduce KV data movement, memory access, and computational complexity by adopting low-rank sparse attention. To the best of our knowledge, it is a **state-of-the-art approach** that achieves **substantial KV-cache compression while maintaining high accuracy**, setting it apart from prior methods such as PALU, LoKI, StreamingLLM, QUEST, DS, Hshare, and ShadowKV, as shown in Table 1 of our paper.

Furthermore, we provide a clear and concise overview of the changes and clarifications made during the rebuttal process, addressing each concern raised.

### 1. Consolidated experimental evidence (long-context & sparse baselines)
- As requested by reviewer YyV8. On RULER with Llama-3.1-8B (128k), SALS-25% is **−0.79 percentage points vs. FP16**, confirming stable accuracy at truly long contexts (Reviewer YyV8 — Final Comment).
- As requested by reviewer BamQ. On LongBench, SALS attains **higher accuracy and faster attention-operator latency** than QUEST/H2O under identical settings (Reviewer BamQ — Table R1).

### 2. More extensive ablation studies
- As requested by reviewer CDzv. Joint-head vs. per-head projections: joint-head consistently **outperforms** per-head on **overlap score** and **overlap rate** (selection accuracy/coverage) (Reviewer CDzv — Table R3).
- As requested by reviewer v3oz. Value path analysis: low-rank compression is ineffective, while two independent quantizers achieve **similar accuracy**, indicating robustness (Reviewer v3oz — Tables R1–R2).

### 3. Algorithmic complexity and latency analysis
- In response to reviewer YyV8. A unified comparison of KV data movement, memory footprint, and computational complexity (Reviewer YyV8 — Table R1).
- In response to reviewer CDzv. Attention-operator latency linked to measured memory-access behavior, explaining observed speedups (Reviewer CDzv — Tables R1–R2).

### 4. Relation to prior work and advantages
- As requested by reviewer YyV8 (InfiniGen, ShadowKV, LoKI). We clarified the core step differences. Although InfiniGen and ShadowKV both involve low-rank ideas, their factorization method and usage scenario differ from ours; in contrast, SALS uses a PCA-based, joint-head latent space designed to adapt across contexts. For LoKI, our pipeline differs in KV-cache compression and token selection: SALS compresses keys in latent space, whereas LoKI’s current design does not support key-cache compression.
- As requested by reviewer CDzv (KIVI, KVQuant, LoKI). KIVI/KVQuant are quantization approaches (precision reduction), fundamentally different from our low-rank compression (dimensionality reduction); these techniques are orthogonal and can be combined with SALS. We also provided a step-by-step comparison with LoKI, explaining why LoKI cannot perform key-cache compression under its formulation.
- As requested by reviewer v3oz (NSA). Differences between SALS and NSA include whether additional training is required and the token-selection strategy.
- As requested by reviewer BamQ (novelty). Our main innovations relative to prior work (e.g., PALU, LoKI) are: joint-head compression of the KV cache in a latent space and a token-selection mechanism tailored to that space, jointly delivering memory and latency benefits.

We once again express our appreciation for the constructive feedback and valuable suggestions from all reviewers. Your comments have helped us improve the clarity, comprehensiveness, and rigor of our work through the rebuttal discussion and additional experimental results. We hope our responses and new evidence provided during the rebuttal will support a fair and balanced assessment.

Thank you for your time and consideration.

---

### Note · Authors · 2025-08-12

Dear AC, PCs, and Reviewers,

We sincerely thank all reviewers for their constructive feedback and insightful comments.

**Strengths highlighted by reviewers:**
Reviewers consistently noted the clarity of writing, solid motivation, rigorous theoretical support for joint-head projection, and careful empirical validation. The integration of latent-space scoring prior to RoPE to avoid rank inflation, combined with sparse attention, was seen as a fresh contribution. System-level implementation achieves up to 6.4× KV-cache compression, and delivers 1.4–4.5× end-to-end throughput gains over GPT-Fast.

**Addressing major concerns:**
1. **Novelty vs. prior work:** We clarified that while LokI, ShadowKV, and InfiniGen all involve low-rank ideas, their factorization methods and **key cache compression** strategy differ fundamentally. SALS applies PCA-based joint-head projection in the latent space, enabling **key-cache compression** that **Loki's design cannot support**. Furthermore, SALS compresses newly generated key caches during decoding while ShadowKV is **limited to prefill key compression**. Similarly, InfiniGen performs **no key-cache compression** but instead employs low-rank skewed partial weights to identify critical tokens. As such, the contribution of SALS is to implement pre-RoPE **latent-space compression**, while maintaining minimal accuracy loss.
2. **Long-context stability:** Added 128k-token RULER experiments (LLaMA-3.1-8B) showing only 0.79% drop vs FP16, confirming robustness at truly long contexts.
3. **Ablations & mechanism analysis:** We show that different value-quantization schemes yield similar accuracy, indicating robustness to quantizer choice. Joint-head compression also outperforms per-head in selection accuracy and coverage, supporting our design.

4. **Complexity & latency:** Provided unified analysis of KV movement, memory footprint, and operator latency linked to measured memory-access patterns.

**Commitment:**
In the camera-ready version, we will integrate these extended experiments, expand the discussion on hybrid compression strategies, and refine the presentation for broader accessibility.

We appreciate the reviewers’ recognition of our contribution’s novelty, rigor, and practical impact, and believe the new evidence provided further substantiates its significance. We hope this work will inspire further exploration in efficient long-context transformer inference.

**Thank you for your consideration.**

---

### Decision · Program_Chairs · 2025-09-17

**Decision:**

Accept (poster)

**Comment:**

This paper introduces SALS, a method for KV cache compression in LLMs that combines pre-RoPE low-rank projection with sparse token selection in a latent space. The key insight is that applying RoPE inflates the effective rank of key vectors, making compression less effective; by compressing in the pre-RoPE space and using joint-head PCA projections, SALS achieves substantial KV-cache compression and attention operator speedups while maintaining accuracy. The approach is carefully engineered, and the authors provide system-level implementation and evaluations showing up to 6.4× compression, 5.7× operator speedup, and 1.4–4.5× throughput gains on contexts up to 128k tokens.

The main strengths are the clarity of exposition, strong empirical validation (including extensions during rebuttal to RULER at 128k tokens, and comparisons to baselines such as Quest, H2O, and ShadowKV), and theoretical justification for joint-head projection. Reviewers appreciated the careful design and the robustness of results across models.

The weaknesses lie in the perceived incremental novelty. Several reviewers noted that SALS builds upon existing low-rank and sparsity ideas (e.g., Loki, ShadowKV, InfiniGen), and while the pre-RoPE latent-space integration is a meaningful refinement, it may not represent a fundamental theoretical breakthrough. Evaluation breadth was also initially limited, though the rebuttal strengthened this with additional long-context and baseline comparisons. Some questions about interaction with GQA/MQA and value-cache compression were raised, and the authors provided clarifications and ablations showing compatibility and robustness.

Overall, the rebuttal and discussion addressed most concerns to a reasonable degree. While the contribution is more incremental than transformative, the system-level execution, thorough analysis, and empirical results establish SALS as a useful and practical advancement for efficient long-context inference.